# Sparse DETR: Efficient End-to-End Object Detection with Learnable Sparsity

**Byungseok Roh**[1*†]**, JaeWoong Shin**[2*‡]**, Wuhyun Shin**[1*]**, Saehoon Kim**[1]
[1]KakaoBrain
[2]Lunit

{peter.roh,aiden.hsin,sam.kim}@kakaobrain.com
jwoong.shin@lunit.io

## ABSTRACT

DETR is the first end-to-end object detector using a transformer encoder-decoder architecture and demonstrates competitive performance but low computational efficiency on high resolution feature maps. The subsequent work, Deformable DETR, enhances the efficiency of DETR by replacing dense attention with deformable attention, which achieves $10\times$ faster convergence and improved performance. Deformable DETR uses the multiscale feature to ameliorate performance, however, the number of encoder tokens increases by $20\times$ compared to DETR, and the computation cost of the encoder attention remains a bottleneck. In our preliminary experiment, we observe that the detection performance hardly deteriorates even if only a part of the encoder token is updated. Inspired by this observation, we propose `Sparse DETR` that selectively updates only the tokens expected to be referenced by the decoder, thus help the model effectively detect objects. In addition, we show that applying an auxiliary detection loss on the selected tokens in the encoder improves the performance while minimizing computational overhead. We validate that `Sparse DETR` achieves better performance than Deformable DETR even with only 10% encoder tokens on the COCO dataset. Albeit only the encoder tokens are sparsified, the total computation cost decreases by 38% and the frames per second (FPS) increases by 42% compared to Deformable DETR. Code is available at https://github.com/kakaobrain/sparse-detr.

## 1 INTRODUCTION

In recent years, we have witnessed the dramatic advancement and the success of object detection in deep learning. Diverse object detection methods have been proposed, but the existing algorithms that perform positive matching with the ground truth as a heuristic way require non-maximum suppression (NMS) post-processing of near-duplicate predictions. Recently, Carion et al. (2020) has introduced a fully end-to-end detector DETR by eliminating the need for NMS post-processing through a set-based objective. The training objective is designed by employing the Hungarian algorithm that considers both classification and regression costs, and achieves highly competitive performance. However, DETR is unable to use multi-scale features such as feature pyramid networks (Lin et al., 2017), which are commonly used in object detection to improve the detection of small objects. The main reason is increased memory usage and computation by adding Transformer (Vaswani et al., 2017) architecture. As a result, its ability to detect small objects is relatively poor.

To address this problem, Zhu et al. (2021) has proposed a deformable-attention inspired by the deformable convolution (Dai et al., 2017) and reduced the quadratic complexity to linear complexity through key sparsification in the attention module. By using deformable attention, deformable DETR addresses the slow convergence and high complexity issue of DETR, which enables the encoder to use multi-scale features as an input and significantly improves performance on detecting small objects. However, using the multi-scale features as an encoder input increases the number of tokens to be processed by about 20 times. Eventually, despite efficient computation for the same

---

*Equal contribution. †Corresponding author. ‡Work is done during an internship at KakaoBrain.

token length, the overall complexity increases back again, making the model inference slower even than vanilla DETR.

In general, natural images often contain large background regions irrelevant to the objects of interest, and accordingly, in end-to-end detectors, the tokens corresponding to the background also occupy a significant portion. In addition, the importance of each regional feature is not identical, which has been proven by the two-stage detectors successfully do their job by focusing only on the foreground. It suggests that there exists considerable regional redundancy that can be reduced in the detection tasks and seeking to devise an efficient detector focusing on the salient regions is a necessary and natural direction. In our preliminary experiments, we observe the following: (a) during inference of a fully-converged Deformable DETR model on the COCO validation dataset, the encoder tokens referenced by the decoder account for only about 45% of the total, and (b) retraining a new detector while updating only the encoder tokens preferred by the decoder from another fully-trained detector, barely suffers performance loss(0.1 AP degradation). See *Appendix A.9* for the details.

Inspired by this observation, we propose a learnable decoder cross-attention map predictor to sparsify encoder tokens. In the existing methods (Carion et al., 2020; Zhu et al., 2021), the encoder takes all the tokens, i.e. the backbone features combined with corresponding positional embeddings, as input without discrimination. Meanwhile, our approach distinguishes encoder tokens to be referenced later in the decoder and considers only those tokens in self-attention. Therefore, this can significantly reduce the number of encoder tokens involved in the computation and reduce the total computational cost. We further propose the encoder auxiliary loss for selected encoder tokens to improve detection performance while minimizing computational overhead. The proposed auxiliary loss not only improves performance, but also allows training a larger number of encoder layers.

Extensive experiments on the COCO 2017 benchmark (Lin et al., 2014) demonstrate that Sparse DETR effectively reduces computational cost while achieving better detection performance. Without bells and whistles, Sparse DETR using Swin-T (Liu et al., 2021) backbone achieves 48.2 AP with 38% reduction of the entire computational cost compared to the 48.0 AP baseline and 49.2 AP with 23% reduction. In the case of the experiment that achieves 48.2 AP using only 10% of encoder tokens, the computational cost of the transformer encoder block is reduced by approximately 82%.

We summarize our contributions as follows:

- We propose *encoder token sparsification* method for an efficient end-to-end object detector, by which we lighten the attention complexity in the encoder. This efficiency enables stacking more encoder layers than Deformable DETR, leading to performance improvement within the same computational budget.
- We propose two novel sparsification criteria to sample the informative subset from the entire token set: *Objectness Score (OS)* and *Decoder cross-Attention Map (DAM)*. Based on the decoder cross-attention map criterion, the sparsified model preserves detection performance even when using only 10% of the whole tokens.
- We adopt an *encoder auxiliary loss* only for the selected tokens. This additional loss not only stabilizes the learning process, but also greatly improves performance, with only marginally increased training time.

## 2 RELATED WORK

**Efficient computation in vision transformers.** It is a well-known problem that the attention computation in Transformers incurs the high time and memory complexity. The vision transformers need to digest even bigger token sets as input so that a large body of works (Parmar et al., 2018; Child et al., 2019a; Ho et al., 2019; Wang et al., 2020; Katharopoulos et al., 2020; Choromanski et al., 2021; Kitaev et al., 2020) has been proposed lightweight attention mechanisms for them. Most of those works shed light on the complexity that resides only in a single-scale attention module, which hinders direct extension to the multi-scale features generally required in object detection.

One of the promising approaches for the lighter transformer attention is input-dependent *token* sparsification. DynamicViT (Rao et al., 2021) and IA-RED[2] (Pan et al., 2021), similar to our work, both propose jointly-learned token selectors generating the sparsity patterns to be overlaid on the input tokens. Those approaches mainly focus on sparsifying a backbone network evaluated on the classification tasks, while our interest lies in a sparse encoder of the end-to-end object detectors.

On the other hand, there has been a line of works sharing the spirit with ours in that they aim at sparse transformers in the DETR-based framework. Deformable DETR (Zhu et al., 2021) conducts sparse attention computation by sampling only a fraction of the entire key set with learnable 2-d offsets, which enables to use multi-scale feature maps with a reasonable computational cost. It can be viewed as a *key* sparsification method but with dense queries, while our approach further reduces the *query* set pursuing even more sparsity. PnP-DETR (Wang et al., 2021) shortens the token length of the transformer encoder by introducing the Polling and Pull (PnP) module to sample the foreground tokens and condense the background tokens into a smaller set. However, their method cannot naively be integrated with Deformable DETR, since their sparsification breaks the 2d spatial structure of the token set assumed in the deformable attention. On the contrary, Sparse DETR preserves the 2d sample space of the set and can be seamlessly combined with the deformable attention, which facilitates handling the multi-scale features. Thus, our approach gets benefits from *both* the deformable *key* sampling and the proposed *query* sparsification. Most of all, we propose explicit objectives for the token selection network, whereas the aforementioned works have no explicit objective implying their beliefs in a *good* selection strategy, merely relying on the final detection objective.

**Auxiliary Loss.** Auxiliary loss (Lee et al., 2015; Szegedy et al., 2015) is widely adopted to deliver gradients to the early layers of deep networks. DETR variants employ auxiliary Hungarian matching objectives at the end of every decoder layer with extra FFN heads so that each decoder layer directly learns to detect the correct number of objects out of the decoder's outputs. Unlike the decoder's object queries whose number is relatively small(e.g. 300), the number of encoder's tokens has much larger scales when using multi-scale features. Thus, extending the layerwise auxiliary loss to the multi-scale encoder increases the training time cost by feeding too many tokens to the attached FFN heads. In Sparse DETR, thanks to the sparsity already induced in the encoder, we can instantly economize that cost while enjoying the auxiliary gradients in a wider range of intermediate layers.

## 3 APPROACH

In this section, we present our main contributions: (a) formulating a generalized saliency-based token sparsification scheme for the encoder, (b) proposing the effective saliency criteria with which that scheme can practically work, and (c) employing the encoder auxiliary losses and the top-$k$ decoder query selection to improve the performance. Before describing the details, we revisit the key components of DETR (Carion et al., 2020) and Deformable DETR (Zhu et al., 2021).

### 3.1 PRELIMINARY

**DETR.** DETR takes the flattened spatial feature map $\mathbf{x}_{\text{feat}} \in \mathbb{R}^{N \times D}$ from a backbone network into the transformer encoder, where $N$ denotes the number of tokens (i.e. features) and $D$ denotes token dimension. The encoder iteratively updates $\mathbf{x}_{\text{feat}}$ by several vanilla self-attention modules. Then, the transformer decoder takes both the refined encoder tokens (i.e. encoder output) and $M$ learnable object queries $\{q_i\}_{i=1\cdots M}$ as inputs and predicts a tuple of a class score $\mathbf{c} \in [0, 1]^C$ and a bounding box $\mathbf{b} \in [0, 1]^4$ for each object query $q_i$, denoted as $\{\hat{\mathbf{y}}_i\} = \{(\mathbf{c}_i, \mathbf{b}_i)\}$, where $C$ denotes the number of classes. All components including the backbone network are jointly trained by performing the bipartite matching between the ground truth $\{\mathbf{y}_i\}$ and predictions $\{\hat{\mathbf{y}}_i\}$.

**Deformable DETR.** Deformable DETR replaces the vanilla dense attention, which is the main computational bottleneck in DETR, with a deformable attention module. This significantly reduces the computational cost and improves the convergence. Suppose that we have the same size of a set of queries (denoted as $\Omega_q$) and a set of keys (denoted as $\Omega_k$), which means $|\Omega_q| = |\Omega_k| = N$. The conventional dense attention computes the attention weight $A_{qk}$ for every pair $\{(q, k) : q \in \Omega_q, k \in \Omega_k\}$, resulting in quadratic complexity with respect to $N$. Deformable attention reduces this quadratic complexity into the linear one by only considering relevant keys for each query. Specifically, deformable attention computes attention weight $A_{qk}$ for all queries and a small set of keys: $\{(q, k) : q \in \Omega_q, k \in \Omega_{qk}\}$, where $\Omega_{qk} \subset \Omega_k$ and $|\Omega_{qk}| = K \ll N$.

Due to this key sparsification, Deformable DETR is able to use the multi-scale features of the backbone network, improving the detection performance of small objects significantly. Paradoxically, using the multi-scale feature increases the number of tokens in the transformer encoder by about $20\times$

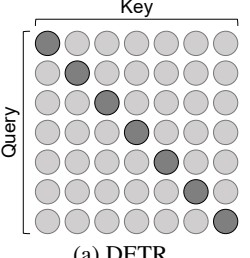
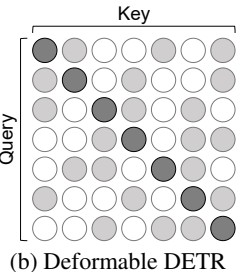
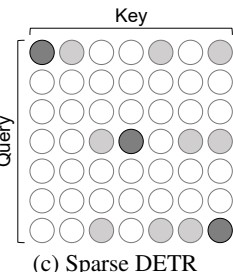

(a) DETR         (b) Deformable DETR         (c) Sparse DETR

Figure 1: **Attention complexity**. The circles in the square matrix represent the attention between keys and queries. The gray/white circles correspond to preserved/removed connection respectively, and darker gray on the diagonal positions means where the token attends to itself. (a) Dense attention in DETR takes quadratic complexity. (b) Deformable DETR uses key sparsification, thus takes linear complexity. (c) Sparse DETR further uses query sparsification. Attention in Sparse DETR also takes linear complexity, but is much lighter than Deformable DETR's.

compared to DETR, making that the encoder becomes the computational bottleneck of deformable DETR. This motivates us to develop a sparsification method to reduce the number of tokens in the encoder aggressively, which is described in the next sections.

### 3.2 ENCODER TOKEN SPARSIFICATION

In this section, we introduce our token sparsification scheme that the encoder module selectively refines a small number of encoder tokens. This encoder token subset is obtained from the backbone feature map $\mathbf{x}_{\text{feat}}$ with a certain criterion, which is described in the subsequent section. For features that are not updated in this process, the values of $\mathbf{x}_{\text{feat}}$ are passed through the encoder layers without being changed.

Formally, suppose that we have a scoring network $g : \mathbb{R}^d \to \mathbb{R}$ that measures saliency of each token in $\mathbf{x}_{\text{feat}}$. We then define $\rho$-salient regions $\Omega_s^\rho$ as the top-$\rho\%$ tokens with the highest scores, for a given keeping ratio $\rho$, i.e. $S = |\Omega_s^\rho| = \rho \cdot |\Omega_q| \ll |\Omega_q| = N$. Then, the $i$-th encoder layer updates the features $\mathbf{x}_{i-1}$ by:

$$\mathbf{x}_i^j = \begin{cases} \mathbf{x}_{i-1}^j & j \notin \Omega_s^\rho \\ \text{LN}(\text{FFN}(\mathbf{z}_i^j) + \mathbf{z}_i^j) & j \in \Omega_s^\rho, \text{ where } \mathbf{z}_i^j = \text{LN}(\text{DefAttn}(\mathbf{x}_{i-1}^j, \mathbf{x}_{i-1}) + \mathbf{x}_{i-1}^j), \end{cases} \quad (1)$$

where DefAttn refers to deformable attention, LN to layer normalization (Ba et al., 2016), and FFN to a feed-forward network. Even in the case of unselected tokens, the values are still passed through the encoder layer, so they can be referenced as keys when updating the selected tokens. This means that the unselected tokens can hand over information to the selected tokens without losing the value of themselves while minimizing the computational cost. Here, we use deformable attention for refining tokens in $\Omega_s^\rho$, but the proposed encoder token sparsification is applicable regardless of which attention method the encoder uses.

**Complexity of Attention Modules in Encoder.** Deformable DETR reduces the attention complexity through key sparsification, and we further reduce the attention complexity through query sparsification, as shown in Fig. 1. Conventional dense attention in DETR requires quadratic complexity $O(N^2)$, where $N$ is the query length. Deformable attention requires linear complexity $O(NK)$, where $K \ll N$ is the number of keys for each query. Sparse attention requires only $O(SK)$, where $S \ll N$ is the number of salient encoder queries.

### 3.3 FINDING SALIENT ENCODER TOKENS

In this section, we introduce how to find a salient token set $\Omega_s^\rho$ from a backbone feature $\mathbf{x}_{\text{feat}}$. We propose a method for determining saliency using a cross attention map from the transformer decoder. Before presenting our approach, we first discuss a simple yet effective method based on the objectness scores obtained from a separate detection head. The limitation of this simple approach motivates us to develop an advanced one, which is described in the following paragraph.

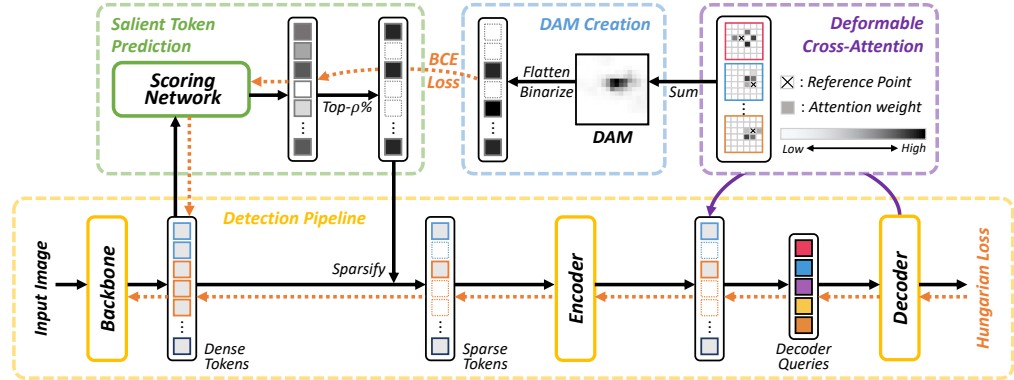

Figure 2: Illustration on how to learn a scoring network by predicting binarized Decoder cross-Attention Map (DAM), where a dashed orange arrow means a backpropagation path. The bottom box shows the forward/backward passes in Sparse DETR, and the top boxes present how to construct DAM for learning the scoring network. See *Appendix A.1* for implementation details of scoring net.

**Objectness Score.**  Measuring objectness per each input token (i.e. feature $\mathbf{x}_{\text{feat}}$) of encoder is very natural to determine which ones from a backbone feature should be further updated in the transformer encoder. It is widely known that feature map from a pretrained backbone network is able to find the saliency of objects, which is why the region proposal network (RPN) has been successfully adopted in many object detectors (Ren et al., 2015; Dai et al., 2016; He et al., 2017). Inspired by this observation, we introduce an additional detection head and Hungarian loss to the backbone feature map, where the structure of the newly added head is the same as the one of the final detection head in the decoder. Then, we can select the top-$\rho\%$ encoder tokens with the highest class scores as a salient token set $\Omega_s^\rho$. This approach is effective to sparsify encoder tokens, but we believe that it is sub-optimal to the transformer decoder, because the selected encoder tokens from the separate detection head are not explicitly considered for the decoder.

**Decoder Cross-Attention Map.**  We consider another approach to select a subset of encoder tokens that are highly relevant to the decoder in a more explicit manner. We observe that the cross-attention maps from the transformer decoder could be used for measuring the saliency, because the decoder gradually attends to a subset of encoder output tokens that are favorable to detect objects as training continues. Motivated by this, we introduce a scoring network that predicts a *pseudo ground-truth* of the saliency defined by decoder cross-attention maps, and use it to determine which encoder tokens should be further refined on the fly. Fig. 2 summarizes how to train a scoring network, and details are presented below.

To determine the saliency of each input token of encoder $\mathbf{x}_{\text{feat}}$, we have to aggregate the decoder cross-attentions between all object queries and the encoder output. This procedure produces a single map of the same size as the feature map from the backbone, which is defined as Decoder cross-Attention Map (DAM). In the case of the dense attention, DAM can be easily obtained by summing up attention maps from every decoder layer. In case of deformable attention, for each encoder token, the corresponding value of DAM can be obtained by accumulating the attention weights of decoder object queries whose attention offsets are directed toward the encoder output tokens. Refer to the *Appendix A.2* for the details in the DAM creation.

To train the scoring network, we binarize DAM so that the top-$\rho\%$ (by attention weights) of encoder tokens is only retained. This is because our goal is to find a small subset of encoder tokens that the decoder references the most, rather than precisely predicting how much each encoder token will be referenced by the decoder. This binarized DAM implies the one-hot target that indicates whether each encoder token is included in the top-$\rho\%$ most referenced encoder tokens. Then, we consider a 4-layer scoring network $g$ to predict how likely a given encoder token is included in the top-$\rho$ % most referenced tokens, and the network is trained by minimizing the binary cross entropy (BCE) loss between the binarized DAM and prediction:

$$\mathcal{L}_{dam} = -\frac{1}{N}\sum_{i=1}^{N}\text{BCE}(g(\mathbf{x}_{\text{feat}})_i, \text{DAM}_i^{\text{bin}}), \qquad (2)$$

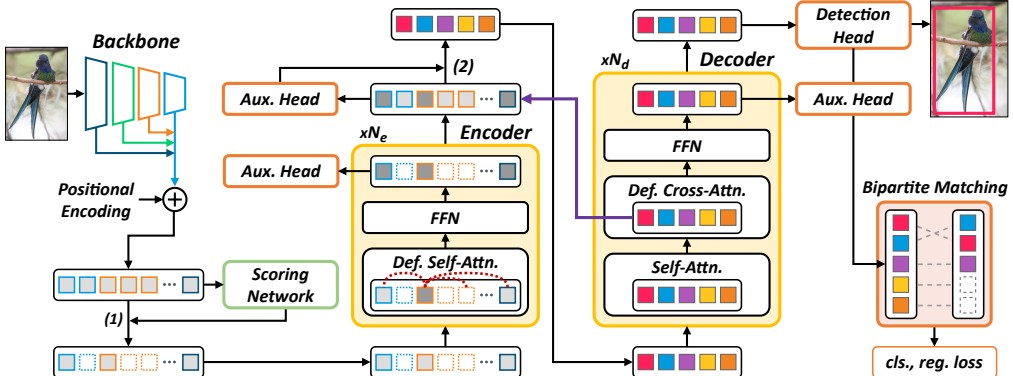

Figure 3: **Sparse DETR architecture**. Sparse DETR introduces three additional components: (a) the scoring network, (b) auxiliary heads in the encoder, and (c) the auxiliary head to select the top-$k$ tokens for the decoder. Sparse DETR measures the saliency of encoder tokens by using the scoring network, and selects the top-$\rho$% tokens, which is referred to as (1) in the diagram. After refining only the selected tokens in the encoder blocks, the auxiliary head selects the top-$k$ tokens from the encoder output, which is served as the decoder object queries. This process is referred to as (2) in the diagram. In addition, we remark that additional auxiliary heads in each encoder block play a key role in achieving improved performance. Only sparsified encoder tokens are passed to the encoder auxiliary heads for efficiency. All auxiliary heads in the encoder and decoder are trained with a Hungarian loss as described in Deformable DETR (Zhu et al., 2021).

where $\text{DAM}_i^{\text{bin}}$ means the binarized DAM value of the $i$th encoder token.

One may say that since DAM in the early phase of training is not accurate, pruning out the encoder tokens based on the result in the decoder degrades the final performance or hurts the convergence. However, we empirically observe that the optimization is very stable even in the early phase of training, and achieves better performance compared to the method based on objectness score. We describe detailed comparisons in the experiments section.

## 3.4 ADDITIONAL COMPONENTS

In this section, we introduce two additional components: (a) auxiliary losses on the encoder tokens and (b) top-$k$ decoder queries selection. We empirically observe that these greatly help improve the final performance and stabilize the optimization. The overall architecture of Sparse DETR including these components is depicted in Fig. 3.

**Encoder Auxiliary Loss.** In DETR variants, auxiliary detection heads are attached to decoder layers, but not to encoder layers. Due to a significantly larger number of encoder tokens (about 18k tokens) compared to decoder tokens (about 300), encoder auxiliary heads will heavily increase the computational cost. In Sparse DETR, however, only part of encoder tokens are refined by the encoder, and adding auxiliary heads only for sparsified encoder tokens is not a big burden.

We empirically observe that applying an auxiliary detection head along with Hungarian loss on the selected tokens stabilizes the convergence of deeper encoders by alleviating the vanishing gradient issue and even improves the detection performance. We conjecture that, following the analysis in Sun et al. (2021), applying Hungarian loss at the intermediate layers helps distinguish the confusing features in the encoder, which contributes to the detection performance in the final head.

**Top-$k$ Decoder Queries.** In DETR and Deformable DETR, decoder queries are given by only learnable object queries or with predicted reference points via another head after the encoder. In Efficient DETR (Yao et al., 2021), decoder takes a part of encoder output as input, similar to RoI Pooling (Ren et al., 2015). Here, an auxiliary detection head is attached to the encoder output $\mathbf{x}_{\text{enc}}$ and the head calculates the objectness (class) score of each encoder output. Based on the score, the top-$k$ encoder outputs are passed as decoder queries, similar to objectness score-based encoder token sparsification. Since this outperforms the methods based on learnable object queries or the two-stage scheme, we include this top-$k$ decoder query selection in our final architecture.

Table 1: **Detection results of Sparse DETR on COCO 2017 val set.** Top-$k$ & BBR denotes that we sample the top-$k$ object queries instead of using the learned object queries (Yao et al., 2021), and perform bounding box refinement in the decoder block (Zhu et al., 2021), respectively. Note that the proposed encoder auxiliary loss is only applied to Sparse DETR. FLOPs and FPS are measured in the same way as used in Zhu et al. (2021). The results marked by †, ‡ are the reported ones from Zhu et al. (2021) and Wang et al. (2021), respectively.

| Method | Epochs | Keeping ratio ($\rho$) | Top-$k$ & BBR | AP | $AP_{50}$ | $AP_{75}$ | $AP_S$ | $AP_M$ | $AP_L$ | params | FLOPs | FPS |
|---|---|---|---|---|---|---|---|---|---|---|---|---|
| *ResNet-50 backbone:* | | | | | | | | | | | | |
| F-RCNN-FPN[†] | 109 | N/A | | 42.0 | 62.1 | 45.5 | 26.6 | 45.4 | 53.4 | 42M | 180G | 26 |
| DETR[†] | 500 | 100% | | 42.0 | 62.4 | 44.2 | 20.5 | 45.8 | 61.1 | 41M | 86G | 28 |
| DETR-DC5[†] | 500 | 100% | | 43.3 | 63.1 | 45.9 | 22.5 | 47.3 | 61.1 | 41M | 187G | 12 |
| PnP-DETR[‡] | 500 | 33% | | 41.1 | 61.5 | 43.7 | 20.8 | 44.6 | 60.0 | - | - | - |
| | 500 | 50% | | 41.8 | 62.1 | 44.4 | 21.2 | 45.3 | 60.8 | - | - | - |
| PnP-DETR-DC5[‡] | 500 | 33% | | 42.7 | 62.8 | 45.1 | 22.4 | 46.2 | 60 | - | - | - |
| | 500 | 50% | | 43.1 | 63.4 | 45.3 | 22.7 | 46.5 | 61.1 | - | - | - |
| Deformable-DETR | 50 | 100% | | 43.9 | 62.8 | 47.8 | 26.1 | 47.4 | 58.0 | 40M | 173G | 19.1 |
| | 50 | 100% | ✓ | 46.0 | 65.2 | 49.8 | 28.2 | 49.1 | 61.0 | 41M | 177G | 18.2 |
| **Sparse-DETR** | 50 | 10% | ✓ | 45.3 | 65.8 | 49.3 | 28.4 | 48.3 | 60.1 | 41M | 105G | 25.3 |
| | 50 | 20% | ✓ | 45.6 | 65.8 | 49.6 | 28.5 | 48.6 | 60.4 | 41M | 113G | 24.8 |
| | 50 | 30% | ✓ | 46.0 | 65.9 | 49.7 | 29.1 | 49.1 | 60.6 | 41M | 121G | 23.2 |
| | 50 | 40% | ✓ | 46.2 | 66.0 | 50.3 | 28.7 | 49.0 | 61.4 | 41M | 128G | 21.8 |
| | 50 | 50% | ✓ | 46.3 | 66.0 | 50.1 | 29.0 | 49.5 | 60.8 | 41M | 136G | 20.5 |
| *Swin-T backbone:* | | | | | | | | | | | | |
| DETR | 500 | 100% | | 45.4 | 66.2 | 48.1 | 22.9 | 49.5 | 65.9 | 45M | 92G | 26.8 |
| Deformable-DETR | 50 | 100% | | 45.7 | 65.3 | 49.9 | 26.9 | 49.4 | 61.2 | 40M | 180G | 15.9 |
| | 50 | 100% | ✓ | 48.0 | 68.0 | 52.0 | 30.3 | 51.4 | 63.7 | 41M | 185G | 15.4 |
| **Sparse-DETR** | 50 | 10% | ✓ | 48.2 | 69.2 | 52.3 | 29.8 | 51.2 | 64.5 | 41M | 113G | 21.2 |
| | 50 | 20% | ✓ | 48.8 | 69.4 | 53.0 | 30.4 | 51.9 | 64.8 | 41M | 121G | 20.0 |
| | 50 | 30% | ✓ | 49.1 | 69.5 | 53.5 | 31.4 | 52.5 | 65.1 | 41M | 129G | 18.9 |
| | 50 | 40% | ✓ | 49.2 | 69.5 | 53.5 | 31.4 | 52.9 | 64.8 | 41M | 136G | 18.0 |
| | 50 | 50% | ✓ | 49.3 | 69.5 | 53.3 | 32.0 | 52.7 | 64.9 | 41M | 144G | 17.2 |

## 4 EXPERIMENTS

We compare Sparse DETR with the conventional object detectors, including the recently proposed ones in the DETR family. In addition, we conduct an ablation study to support our claims in Section 3, presenting the performance comparison between token selection criteria (OS vs. DAM), the effectiveness of the encoder auxiliary loss, and the dynamic sparsification during inference.

**Implementation Details.** We use ResNet-50 (He et al., 2016) and Swin Transformer (Liu et al., 2021) as pre-trained backbone networks, where Swin Transformer is one of the state-of-the-art architecture in the ViT family. We stack 6 encoder and 6 decoder layers, each with an auxiliary head at the end. We train the model on a $4\times$V100 GPU machine with a total batch size of 16, for 50 epochs, where the initial learning rate is 0.0002 and decayed by 1/10 at the 40 epoch. Unless otherwise specified, we use the same hyperparameters as in Deformable DETR.

### 4.1 COMPARISON WITH OBJECT DETECTION BASELINES

**Baselines.** We compare Sparse DETR with Faster-RCNN with FPN (Lin et al., 2017), DETR (Carion et al., 2020), Deformable DETR (Zhu et al., 2021), and PnP DETR (Wang et al., 2021). We also compare with DETR and Deformable DETR that uses Swin-Tiny (Liu et al., 2021) as a backbone. Here, for brevity, we denote Deformable DETR with the top-$k$ object query selection and bounding box refinement, as Deformable DETR+. In Sparse DETR, encoder tokens are sparsified with keeping ratios of 10%, 20%, 30%, 40%, and 50%, using DAM criterion. We demonstrate detection performance and inference costs on COCO `val2017` dataset.

**Result.** Table 1 shows the results of Sparse DETR and the other baselines on COCO `val2017` set. Remarkably, on the ResNet-50 backbone, Sparse DETR with a keeping ratio over 30% outper-

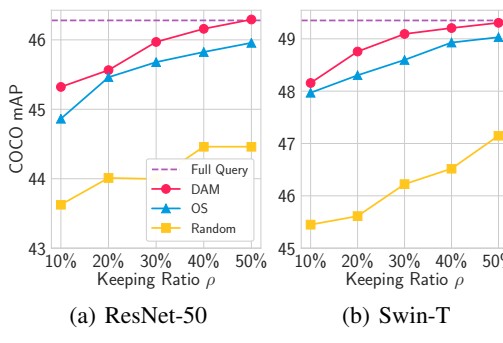

(a) ResNet-50     (b) Swin-T

Figure 4: **Selection criteria.** Comparison of the performance with respect to encoder token selection criteria for different backbones.

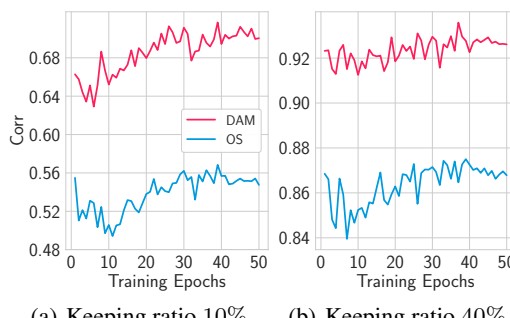

(a) Keeping ratio $10\%$    (b) Keeping ratio $40\%$

Figure 5: **Correlation graph.** Correlation graphs of OS and DAM during training.

forms all the baselines while minimizing the computational cost. Even with the keeping ratio reduced down to 10%, Sparse DETR still performs better than most baselines except for Deformable DETR+. More surprisingly, on the Swin-T backbone, Sparse DETR only with the keeping ratio 10% outperforms all the baselines with no exception, while improving FPS by $38\%$ compared to Deformable DETR+.

Remark that, compared to the most competitive baseline, Deformable DETR+, the improvement in $AP_L$ is relatively noticeable on the Swin-T backbone even under the extreme sparsity of 10%, while the performance gap on the ResNet-50 backbone comes evenly from different sizes of objects. We conjecture that it is because a single token in Swin-T can hold a wider region of information than the one in ResNet-50, so even if we aggressively sparsify the encoder token, the network seems to have enough information to detect objects.

## 4.2    Comparison Between token Selection Criteria

**Baselines.** To verify the benefits of the proposed saliency criteria, we compare three token sparsification criteria: random, Objectness Score (OS), and Decoder cross-Attention Map (DAM). The random baseline samples a fixed ratio of arbitrary tokens. Note that the proposed encoder auxiliary loss is applied for all the methods.

**Result.** As illustrated in Fig. 4, the random strategy suffers noticeable performance degradation. On the other hand, the DAM-based model outperforms the OS-based model at every ratio and almost catches up with the non-sparse baseline when using 50% of encoder tokens. See the *Appendix A.4* for detailed results of this experiment.

To analyze the reason that DAM-based model outperforms its counterpart, we measure the overlap between the encoder tokens referred by the decoder and the tokens refined by the encoder. As a metric, we compute a scalar correlation $Corr$ as:

$$Corr := \frac{\sum_{x \in \Omega_D \cap \Omega_s^\rho} \mathrm{DAM}_x}{\sum_{x \in \Omega_D} \mathrm{DAM}_x},$$

where $\Omega_D$ is the encoder token set referred by the decoder and $\mathrm{DAM}_x$ is the DAM-value corresponding to token $x$. This $Corr$ metric indicates the ratio of tokens polished by the encoder among the tokens referred by the decoder.

Fig. 5 demonstrates that $Corr$ of DAM-based model rises higher than that of OS-based model. This result implies that DAM-based model is a more suitable sparsification method for the decoder, because the tokens referenced by the decoder are explicitly refined in the encoder, which achieves better detection performance. See the *Appendix A.4* for detailed results of this experiment.

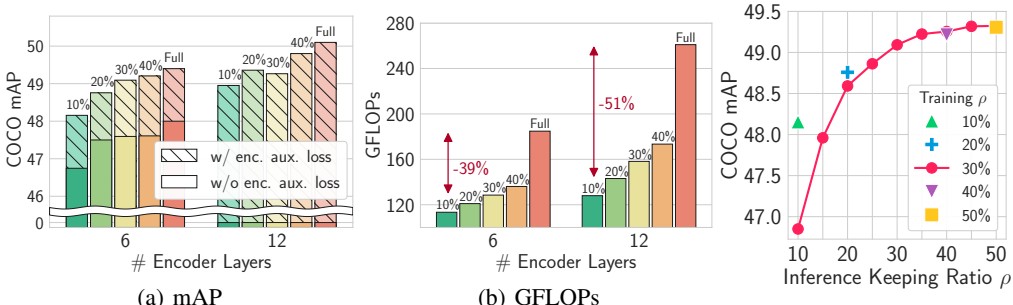

Figure 6: **Ablation of # encoder layers**.    Figure 7: **Dynamic sparsification**.

(a) mAP    (b) GFLOPs

### 4.3 EFFECTIVENESS OF THE ENCODER AUXILIARY LOSS

Owing to the sparsified token set in our model, we can apply the auxiliary loss to the encoder layers without sacrificing too much computational cost. Apart from improved efficiency and performance, we find another benefit of the encoder auxiliary loss that allows us to safely stack more encoder layers without failing to converge.

As shown in Fig. 6, the encoder auxiliary loss not only enhances detection performance, but also consistently increases detection performance as the encoder layers doubled to 12. However, we observe that the training without its assistance utterly fails when using 12 encoder layers. We argue that gradient propagated through decoder cross-attention vanishes as we stack more encoder layers, thus intermediate gradients from the auxiliary loss are required. The observations reported in *Appendix A.5* supports this assertion and *Appendix A.6* details the results of Fig. 6.

### 4.4 DYNAMIC SPARSIFICATION FOR INFERENCE STAGE

To deploy the models in various hardware conditions of real-world applications, one often should retrain them at different scales according to the performance-computation trade-off required. We evaluate if our model trained with a fixed sparsity can adapt well to dynamic sparsity at inference time to check out Sparse DETR can avoid that hassle. Figure 7 shows the performance under the varied keeping ratio ($\rho$) during inference when the model trained using the Swin-T backbone and 30% of encoder tokens with the DAM-based method. When the inference keeping ratio is small, the performance of dynamic sparsification is slightly degraded, but the overall performance is satisfactory at various keeping ratios given that only a single model is used.

PnP DETR introduces *dynamic ratio training* to achieve similar performance to the fixed keeping ratio counterpart. However, without the additional trick, it suffers significant performance degradation, for instance, 5.0 AP drop when training/inference keeping ratio is 0.33/0.5, despite the increased number of encoder tokens. On the contrary, Sparse DETR achieves 0.2 AP improvement in a similar condition where the training/inference keeping ratio is 0.3/0.5. To conclude, our method shows better robustness compared to PnP DETR without further treatment, showing a greater potential of dynamic adaptation to different hardware environments. Note that any technique such as dynamic ratio training is orthogonal to our method and introducing it may bring even more robustness.

## 5 CONCLUSION

In this paper, we have presented encoder token sparsification algorithm that lowers the computational cost of the encoder, which is a computational bottleneck in the DETR and Deformable DETR. By doing so, the proposed Sparse DETR architecture outperforms the Deformable DETR even when using only 10 % of the encoder token, and decreases the overall computation by 38%, and increases the FPS by 42% compared to the Deformable DETR. We hope that our proposed method will provide insights to effectively detect objects in the transformer structure in the future.

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

## A APPENDIX

### A.1 IMPLEMENTATION DETAILS OF THE SCORING NETWORK

The scoring network is consists of 4 linear layers where Layer Normalization (Ba et al., 2016) comes before the first layer and every layer except for the last one is followed by GELU (Hendrycks & Gimpel, 2016) activation. The output dimension of the 1st layer is 256 and halved to 128 and 64 at the 2nd and 3rd layers. The last layer outputs 1-d logit for the BCE loss. Since the network locally processes the input tokens in a token-wise manner, the final decisions may overlook global statistics without additional treatment. To remedy this issue, we set aside half of the output dimension of the first layer as a global feature, and average them across the whole token set, then concatenate it with each of the remained local features to maintain the original dimension. We also exclude the tokens that correspond to the zero-padded area when selecting top-$\rho$% scores, thereby we can prevent those tokens from participating in Hungarian matching process and getting meaningless gradients from the detection objective.

### A.2 DAM CREATION IN DEFORMABLE ATTENTION

As attention offset calculated in deformable attention is a fractional position, deformable attention uses bilinear interpolation to get values. Thus, we also use bilinear interpolation to obtain DAM.

Assume that, one of the attention offsets, weights and the reference point of decoder object query $q$ is calculated as $p$, $A$ and $r$, respectively. Then, deformable attention takes values as:

$$\sum_{x} A \cdot G(x, r + p) \cdot v(x)$$

, where $x$ enumerates all integral spatial locations in the feature map, $G(\cdot, \cdot)$ is the bilinear interpolation kernel defined as $G(a, b) = \max(0, 1 - |a_x - b_x|) \cdot \max(0, 1 - |a_y - b_y|)$ and $v$ is the values. Similarly, we accumulate DAM-value for location $x$ as:

$$\sum_{(p,A,r)} A \cdot G(x, r + p)$$

. Then, we accumulate DAM over every decoder object query.

### A.3 ALTERNATIVE OBJECTIVES FOR DAM-BASED MODEL

As a training objective of the scoring network using DAM, we can consider other alternatives as long as they can encourage the predicted scores to represent the relative saliency of the encoder tokens. One of the naive alternatives is the *regression loss* by which the scoring network directly predicts the values in DAM. The *ranking loss* can be another choice with which the network focuses more on learning the relativeness rather than estimating the set of salient tokens.

Figure 8 shows the default BCE loss outperforms the alternatives. First, it is well-known that the regression problem is much harder than classification. Furthermore, since the value of DAM changes during training, the regression loss to predict the accurate value is more difficult. In case of the pairwise ranking loss, ranking the DAM elements may also be unstable as DAM gradually evolves. Meanwhile, the BCE loss may reduce those element-level noises down to the set-level in that its binary (keep or drop) targets retain more consistency compared to the exact values or ranks. Refer to Table 2 to see the exact values of the points represented in Figure 8.

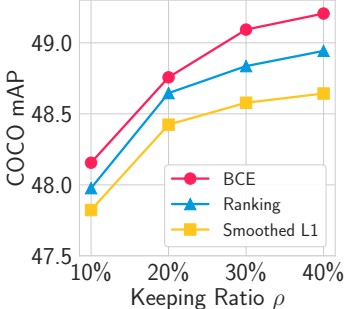

Figure 8: **DAM loss ablation**

Table 2: **Comparision between the alternative objectives for DAM-based scoring network**.

| Loss | Keeping ratio ($\rho$) | AP | $AP_{50}$ | $AP_{75}$ | $AP_S$ | $AP_M$ | $AP_L$ |
|---|---|---|---|---|---|---|---|
| smoothed L1 | 10% | 47.8 | 68.9 | 51.7 | 29.8 | 50.9 | 64.0 |
| | 20% | 48.4 | 69.0 | 52.5 | 31.1 | 51.4 | 64.7 |
| | 30% | 48.6 | 69.0 | 52.6 | 31.1 | 51.9 | 64.7 |
| | 40% | 48.6 | 69.3 | 52.9 | 33.4 | 51.8 | 64.5 |
| ranking | 10% | 48.0 | 69.1 | 52.1 | 29.9 | 51.4 | 64.6 |
| | 20% | 48.7 | 69.5 | 53.0 | 31.1 | 51.8 | 65.1 |
| | 30% | 48.8 | 69.2 | 52.8 | 31.4 | 52.0 | 64.9 |
| | 40% | 48.9 | 69.3 | 53.1 | 31.5 | 52.2 | 64.7 |
| BCE | 10% | 48.2 | 69.2 | 52.3 | 29.8 | 51.2 | 64.5 |
| | 20% | 48.8 | 69.4 | 53.0 | 30.4 | 51.9 | 64.8 |
| | 30% | 49.1 | 69.5 | 53.5 | 31.4 | 52.5 | 65.1 |
| | 40% | 49.2 | 69.5 | 53.5 | 31.4 | 52.9 | 64.8 |

Table 3: **Comparison between token selection criteria**.

| Scoring method | Keeping ratio ($\rho$) | AP | $AP_{50}$ | $AP_{75}$ | $AP_S$ | $AP_M$ | $AP_L$ | params | FLOPs | FPS |
|---|---|---|---|---|---|---|---|---|---|---|
| ***ResNet-50** backbone:* | | | | | | | | | | |
| N/A | 100% | 46.3 | 65.8 | 50.1 | 29.0 | 49.4 | 61.7 | 41M | 177G | 18.2 |
| | 0% | 42.2 | 63.0 | 45.6 | 25.9 | 45.3 | 56.5 | 36M | 91G | 35.0 |
| random | 10% | 43.6 | 64.3 | 47.2 | 26.7 | 46.9 | 58.4 | 41M | 102G | 27.7 |
| | 20% | 44.0 | 64.8 | 47.8 | 27.3 | 47.0 | 58.4 | 41M | 110G | 25.6 |
| | 30% | 44.0 | 64.9 | 47.5 | 27.4 | 47.4 | 58.2 | 41M | 117G | 24.1 |
| | 40% | 44.5 | 65.1 | 48.0 | 27.3 | 47.8 | 59.8 | 41M | 125G | 22.5 |
| | 50% | 44.4 | 64.8 | 48.0 | 27.8 | 47.4 | 59.2 | 41M | 133G | 21.1 |
| OS | 10% | 44.9 | 65.2 | 48.7 | 27.9 | 47.8 | 60.4 | 41M | 106G | 26.6 |
| | 20% | 45.5 | 65.5 | 49.3 | 28.7 | 48.3 | 60.5 | 41M | 114G | 24.7 |
| | 30% | 45.7 | 65.8 | 49.5 | 29.7 | 48.5 | 60.8 | 41M | 121G | 23.2 |
| | 40% | 45.8 | 65.5 | 49.8 | 29.1 | 48.8 | 60.5 | 41M | 129G | 21.8 |
| | 50% | 46.0 | 65.9 | 49.8 | 28.8 | 48.9 | 60.6 | 41M | 136G | 20.6 |
| DAM | 10% | 45.3 | 65.8 | 49.3 | 28.4 | 48.3 | 60.1 | 41M | 105G | 26.5 |
| | 20% | 45.6 | 65.8 | 49.6 | 28.5 | 48.6 | 60.4 | 41M | 113G | 24.8 |
| | 30% | 46.0 | 65.9 | 49.7 | 29.1 | 49.1 | 60.6 | 41M | 121G | 23.2 |
| | 40% | 46.2 | 66.0 | 50.3 | 28.7 | 49.0 | 61.4 | 41M | 128G | 21.8 |
| | 50% | 46.3 | 66.0 | 50.1 | 29.0 | 49.5 | 60.8 | 41M | 136G | 20.5 |
| ***Swin-T** backbone:* | | | | | | | | | | |
| N/A | 100% | 49.4 | 69.4 | 53.5 | 31.9 | 52.6 | 65.1 | 41M | 185G | 15.4 |
| | 0% | 43.7 | 65.8 | 46.9 | 27.0 | 46.7 | 60.0 | 37M | 96G | 26.5 |
| random | 10% | 45.5 | 67.6 | 48.8 | 28.4 | 48.5 | 62.2 | 41M | 110G | 22.1 |
| | 20% | 45.6 | 67.5 | 49.2 | 28.6 | 49.1 | 62.2 | 41M | 118G | 20.8 |
| | 30% | 46.2 | 68.1 | 49.7 | 29.5 | 49.7 | 63.0 | 41M | 125G | 19.7 |
| | 40% | 46.5 | 68.2 | 50.0 | 29.9 | 49.8 | 63.0 | 41M | 133G | 18.7 |
| | 50% | 47.2 | 68.3 | 50.9 | 29.1 | 50.4 | 63.9 | 41M | 141G | 17.7 |
| OS | 10% | 48.0 | 69.1 | 52.1 | 29.9 | 51.1 | 64.4 | 42M | 114G | 21.4 |
| | 20% | 48.3 | 69.1 | 52.5 | 30.4 | 51.6 | 64.2 | 42M | 122G | 20.2 |
| | 30% | 48.6 | 69.2 | 53.0 | 31.0 | 52.0 | 64.6 | 42M | 129G | 18.6 |
| | 40% | 48.9 | 69.4 | 53.1 | 33.0 | 51.9 | 64.5 | 42M | 137G | 18.2 |
| | 50% | 49.0 | 69.2 | 53.5 | 31.2 | 52.4 | 65.0 | 42M | 145G | 17.2 |
| DAM | 10% | 48.2 | 69.2 | 52.3 | 29.8 | 51.2 | 64.5 | 41M | 113G | 21.2 |
| | 20% | 48.8 | 69.4 | 53.0 | 30.4 | 51.9 | 64.8 | 41M | 121G | 20.0 |
| | 30% | 49.1 | 69.5 | 53.5 | 31.4 | 52.5 | 65.1 | 41M | 129G | 18.9 |
| | 40% | 49.2 | 69.5 | 53.5 | 31.4 | 52.9 | 64.8 | 41M | 136G | 18.0 |
| | 50% | 49.3 | 69.5 | 53.3 | 32.0 | 52.7 | 64.9 | 41M | 144G | 17.2 |

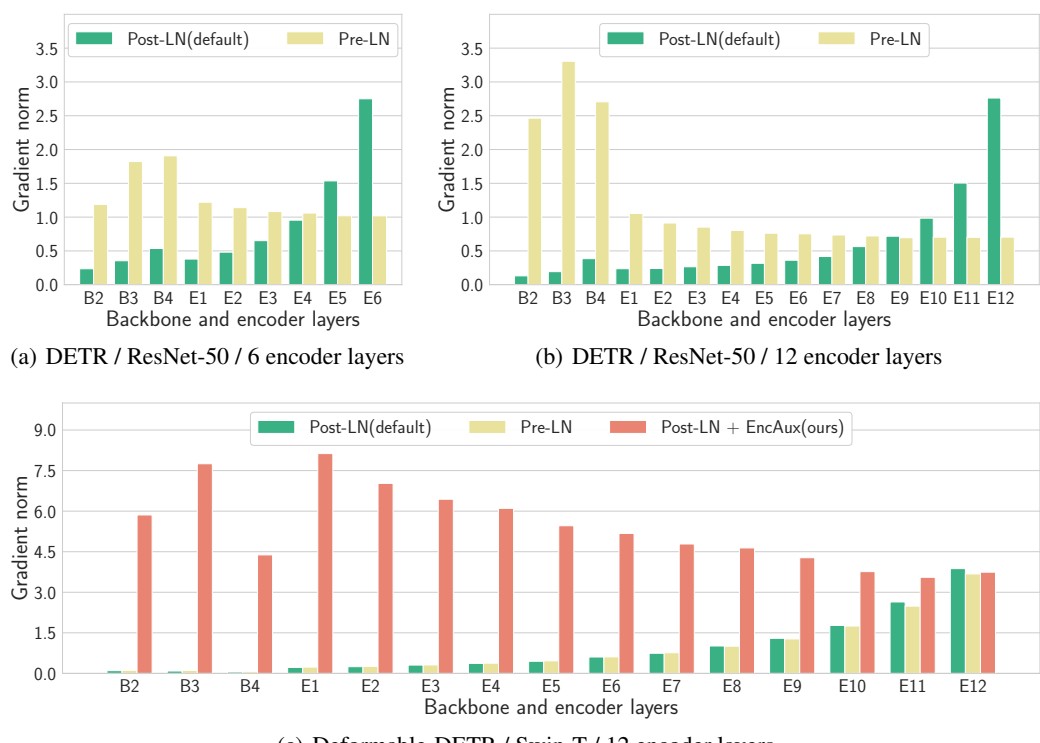

Figure 9: **Layerwise gradient norm in DETR variants.** An observation of the vanishing gradient problem on DETR variants with different backbones by measuring $\ell^2$-norm of gradients in a layerwise manner. The first letter in $x$-axis label represents module name, specifically, 'B' for the backbone and 'E' for the encoder, and the second number represents $i$-th layer in that module. (a), (b): Layerwise gradient norm of DETR with ResNet-50 backbone. With the default settings(Post-LN), the gradient scale generally decreases as more encoder layers are stacked, while the Pre-LN technique preserves gradient magnitude even in deeper early layers. (c) : Layerwise gradient norm of Deformable-DETR with Swin-T backbone. In this case, the Pre-LN fails to prevent vanishing gradient while the encoder auxiliary loss(denoted as Post-LN + EncAux) proposed in this paper effectively resolves this issue.

## A.4 EXPERIMENTAL DETAILS FOR DIFFERENT TOKEN SELECTION CRITERIA

Table 3 contains the specific values used to plot (a) ResNet-50 and (b) Swin-T backbone in Figure 4. Additionally, they also include a lower-bound baseline that has no scoring method with keeping ratio 0%, meaning that the entire encoder block is removed and the backbone features are directly passed to the decoder. Even with the lowest keeping ratio 10%, all the scoring methods including random criterion outperform this lower-bound baseline. Note that all experiments reported in Table 3 except for the keeping ratio 0% use the encoder auxiliary loss for training.

## A.5 VANISHING GRADIENT PROBLEM IN THE DEEP END-TO-END DETECTORS

As shown in Section 4.2 in Carion et al. (2020), they observe that the performance of DETR gradually improves with more encoder layers. To reproduce this result, we used the default settings of the official code, but only changed the number of encoder layers. However, we fail to train the DETR model when using more than 9 encoder layers, which is probably due to different hyperparameters from the ones used in their experiments. Interestingly, we also found that the DETR model converges stably with the Pre-LN architecture(Baevski & Auli, 2019; Child et al., 2019b; Wang et al., 2019) that is known to be a better choice than the canonical Post-LN when the number of layers of

Table 4: **Effectiveness of the encoder auxiliary loss using Swin-T**. When the number of encoder layers is more than 9, the model training fails, but if the encoder auxiliary loss is adopted, the model training is feasible regardless of the number of encoder layers, and accuracy is improved.

| # of encoder | Keeping ratio ($\rho$) | Aux. loss | AP | $AP_{50}$ | $AP_{75}$ | $AP_S$ | $AP_M$ | $AP_L$ | params | FLOPs | FPS |
|---|---|---|---|---|---|---|---|---|---|---|---|
|  | 100% |  | 48.0 | 68.0 | 52 | 30.3 | 51.4 | 63.7 | 41M | 185G | 15.4 |
|  | 100% | ✓ | 49.4 | 69.4 | 53.5 | 31.9 | 52.6 | 65.1 | 41M | 185G | 15.4 |
|  | 10% |  | 46.8 | 68.0 | 50.6 | 29.7 | 49.7 | 63.3 | 41M | 113G | 21.2 |
|  | 20% |  | 47.5 | 68.3 | 51.4 | 31.4 | 50.4 | 64.4 | 41M | 121G | 20.0 |
| 6 | 30% |  | 47.6 | 67.9 | 51.4 | 29.9 | 51.1 | 63.9 | 41M | 129G | 18.9 |
|  | 40% |  | 47.6 | 68.2 | 51.5 | 30.3 | 50.8 | 64.0 | 41M | 136G | 18.0 |
|  | 10% | ✓ | 48.2 | 69.2 | 52.3 | 29.8 | 51.2 | 64.5 | 41M | 113G | 21.2 |
|  | 20% | ✓ | 48.8 | 69.4 | 53.0 | 30.4 | 51.9 | 64.8 | 41M | 121G | 20.0 |
|  | 30% | ✓ | 49.1 | 69.5 | 53.5 | 31.4 | 52.5 | 65.1 | 41M | 129G | 18.9 |
|  | 40% | ✓ | 49.2 | 69.5 | 53.5 | 31.4 | 52.9 | 64.8 | 41M | 136G | 18.0 |
| 9 | 100% | ✓ | 49.7 | 69.4 | 54.1 | 32.4 | 52.9 | 65.4 | 44M | 220G | 12.8 |
|  | 100% | ✓ | 50.1 | 69.6 | 54.6 | 32.2 | 53.4 | 65.8 | 46M | 261G | 11.0 |
|  | 10% | ✓ | 49.0 | 69.5 | 53.5 | 31.6 | 52.2 | 65.2 | 46M | 128G | 19.2 |
| 12 | 20% | ✓ | 49.4 | 69.6 | 53.5 | 31.9 | 52.8 | 65.4 | 46M | 143G | 17.5 |
|  | 30% | ✓ | 49.3 | 69.4 | 53.6 | 31.7 | 52.5 | 65.6 | 46M | 158G | 15.6 |
|  | 40% | ✓ | 49.8 | 69.8 | 54.3 | 33.1 | 53.4 | 65.4 | 46M | 173G | 14.6 |

transformer increases. We used the `pre_norm` option the authors already have implemented in their code.

Figure 9(a) and 9(b) illustrate that gradient norm of each layer from bottom to top in 6 and 12 encoder layers when using Post-LN and Pre-LN, respectively. We compute the $\ell^2$-norm of the gradients for all parameters in a particular layer, as if they are concatenated into a single vector. To see training dynamics in the early stage of training, we track the gradients computed on a fixed set of training data and average them over the first 150 steps with a batch size of 2. Note that we applied Pre-LN only to the encoder module for a fair comparison between only the early modules although it could be used in any other transformer modules, e.g. backbone or decoder.

We found that the vanishing gradient issue is generally observed regardless of the encoder size. When we double the size of the encoder, as one can expect, the gradient in the early layers ends up with an even smaller scale, which may have caused the convergence failure. Meanwhile, the Pre-LN technique seems to significantly alleviate this issue even for the deeper encoder by maintaining the gradient scale evenly through the encoder layers and conveying a strong training signal to the backbone layers.

On the other hand, as shown in Figure 9(c), Deformable-DETR suffers from the same problem of vanishing gradient and even the Pre-LN technique does not help in this case. Meanwhile, the encoder auxiliary loss proposed in our paper drastically amplifies the gradient magnitude in the early layers by providing aggressive intermediate objectives for each encoder layer. Note that it also creates good synergy with our main sparsification strategy owing to reduced training cost. We claim that this observation supports our motivation of introducing the encoder auxiliary loss.

## A.6    EXPERIMENTAL DETAILS FOR EFFECTIVENESS OF THE ENCODER AUXILIARY LOSS

Table 4 presents the detailed values of Figure 4. As shown in the Table 4 and discussed in Section A.5, training of a deeper encoder of more than 9 layers fails without the auxiliary loss, but if it is adopted, the convergence becomes feasible as the intermediate gradients provided to the early encoder layers augment the vanishing gradient back-propagated from the decoder module.

Table 5: **Detection results of Sparse DETR with SCRL initialization using ResNet-50**. The same environment and hyperparameters as Experiments section are used, except for initializing the backbone with SCRL (Roh et al., 2021) model. The results marked by § mean that the backbone network is initialized by SCRL instead of the ImageNet (Deng et al., 2009) pre-trained one.

| Method | Keeping ratio ($\rho$) | AP | $AP_{50}$ | $AP_{75}$ | $AP_S$ | $AP_M$ | $AP_L$ | params | FLOPs | FPS |
|---|---|---|---|---|---|---|---|---|---|---|
| **Sparse-DETR** | 10% | 45.3 | 65.8 | 49.3 | 28.4 | 48.3 | 60.1 | 41M | 105G | 25.3 |
| | 20% | 45.6 | 65.8 | 49.6 | 28.5 | 48.6 | 60.4 | 41M | 113G | 24.8 |
| | 30% | 46.0 | 65.9 | 49.7 | 29.1 | 49.1 | 60.6 | 41M | 121G | 23.2 |
| | 40% | 46.2 | 66.0 | 50.3 | 28.7 | 49.0 | 61.4 | 41M | 128G | 21.8 |
| | 50% | 46.3 | 66.0 | 50.1 | 29.0 | 49.5 | 60.8 | 41M | 136G | 20.5 |
| **Sparse-DETR**§ | 10% | 46.9 | 67.2 | 51.0 | 30.2 | 49.7 | 62.3 | 41M | 105G | 25.3 |
| | 20% | 47.3 | 67.1 | 51.4 | 29.7 | 50.3 | 62.7 | 41M | 113G | 24.8 |
| | 30% | 47.4 | 67.3 | 51.4 | 30.1 | 50.5 | 62.4 | 41M | 121G | 23.2 |
| | 40% | 47.7 | 67.4 | 51.6 | 30.0 | 50.8 | 62.9 | 41M | 128G | 21.8 |
| | 50% | 47.9 | 67.5 | 52.1 | 30.5 | 51.2 | 63.2 | 41M | 136G | 20.5 |

Table 6: **Performance of Sparse DETR with Swin-B**. The same environment and hyperparameters as Experiments section are used, except for changing the backbone to a larger scale. Note that Aux. loss means only the ones applied to the encoder layers.

| Backbone | Keeping ratio ($\rho$) | Aux. loss | AP | $AP_{50}$ | $AP_{75}$ | $AP_S$ | $AP_M$ | $AP_L$ | params | FLOPs | FPS |
|---|---|---|---|---|---|---|---|---|---|---|---|
| Swin-T | 100% | | 48.0 | 68.0 | 52.0 | 30.3 | 51.4 | 63.7 | 41M | 185G | 15.4 |
| | 10% | ✓ | 48.2 | 69.2 | 52.3 | 29.8 | 51.2 | 64.5 | 41M | 113G | 21.2 |
| | 20% | ✓ | 48.8 | 69.4 | 53.0 | 30.4 | 51.9 | 64.8 | 41M | 121G | 20.0 |
| | 30% | ✓ | 49.1 | 69.5 | 53.5 | 31.4 | 52.5 | 65.1 | 41M | 129G | 18.9 |
| | 40% | ✓ | 49.2 | 69.5 | 53.5 | 31.4 | 52.9 | 64.8 | 41M | 136G | 18.0 |
| Swin-B | 100% | | 52.5 | 72.9 | 56.9 | 34.7 | 56.5 | 69.6 | 101M | 400G | 7.6 |
| | 10% | ✓ | 52.2 | 73.5 | 57.0 | 34.0 | 56.3 | 70.3 | 101M | 335G | 8.8 |
| | 20% | ✓ | 53.1 | 73.8 | 57.9 | 34.6 | 56.9 | 70.6 | 101M | 343G | 8.6 |
| | 30% | ✓ | 53.2 | 73.7 | 57.7 | 35.3 | 56.8 | 70.8 | 101M | 350G | 8.4 |
| | 40% | ✓ | 53.3 | 73.4 | 58.0 | 36.3 | 57.2 | 70.9 | 101M | 358G | 8.2 |

A.7    EFFECTIVENESS OF USING A DENSE REPRESENTATION AS BACKBONE INITIALIZATION

Recently, many self-supervised learning methods through contrastive learning have been studied, and in particular, methods for obtaining dense representations with better performance for localization downstream tasks such as object detection are in the spotlight. In order to check whether our proposed method is effective even when such dense representation is used, the backbone network is initialized with the SCRL (Roh et al., 2021) model that aims to learn dense representations in a self-supervised way instead of initializing with the ImageNet (Deng et al., 2009) pre-trained one. Just as the SCRL model outperformed the ImageNet pre-trained model in various localization downstream tasks, our proposed method, Sparse DETR, also shows better performance in all keeping ratios($\rho$) without the influence of encoder token sparsification as shown in Table 5.

A.8    USING A LARGER TRANSFORMER-BASED BACKBONE(SWIN-B)

We perform experiments on Sparse DETR with Swin-Base(Liu et al., 2021) backbone to see if our method shows similar efficiency and performance gain even when using a heavier transformer-based backbone. Table 6 illustrates a comparison of COCO detection performance between Swin-T and Swin-B backbone under the varied sparsity. Due to the increased capacity, using Swin-B backbone significantly boosts up the baseline AP up to 52.5 (+4.5) but with 2.4× parameters and 2.1× com-

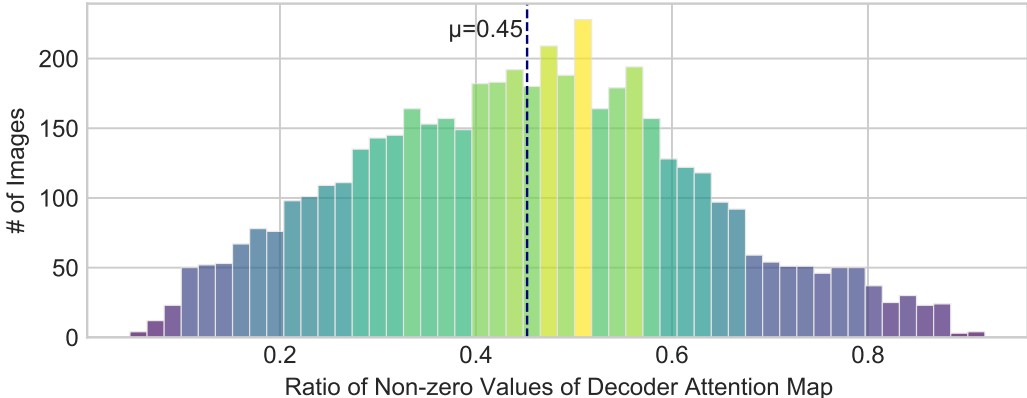

Figure 10: **Distribution of the ratio of non-zero values of DAM on COCO 2017 val set**.

Table 7: **Two-stage encoder token sparsification with a varied keeping ratio**. COCO detection performance when the encoder tokens are sparsified at the later stage with the top-$\rho$% binarized DAMs pre-computed from the former stage. All models are Deformable-DETR+ with Swin-T backbone and the encoder auxiliary loss is not applied. Note that the performance of the 50% model(47.9 AP) hardly degenerates compared to the baseline(48.0 AP).

| Keeping ratio ($\rho$) | AP | $AP_{50}$ | $AP_{75}$ | $AP_S$ | $AP_M$ | $AP_L$ |
|---|---|---|---|---|---|---|
| 100% | 48.0 | 68.0 | 52.0 | 30.3 | 51.4 | 63.7 |
| 10% | 44.0 | 66.0 | 47.2 | 26.9 | 46.8 | 61.0 |
| 20% | 44.9 | 66.3 | 48.3 | 28.2 | 48.2 | 61.4 |
| 30% | 46.5 | 67.3 | 50.2 | 30.9 | 49.7 | 62.3 |
| 40% | 47.3 | 67.9 | 51.3 | 30.7 | 50.7 | 63.4 |
| 50% | 47.9 | 67.8 | 52.0 | 29.8 | 51.4 | 63.7 |

putational cost. With the keeping ratio of 40% and the encoder auxiliary loss, the performance gap remains at a similar level(+4.1). We can also observe consistent performance gains as the keeping ratio gets higher while the increasing gap converges more quickly than Swin-T. It may be because a single visual token of Swin-B can incorporate a wider range of information due to the deeper attention hierarchies and a smaller number of tokens is required to fully represent all the objects in an image. Note that the efficiency of a backbone network is behind the scope of this paper. Our work is orthogonal to the backbone sparsification approaches, e.g. DynamicViT (Rao et al., 2021), and we leave the integration with those works as future work.

## A.9 THE PRELIMINARY EXPERIMENTS: WHY PURSUE A SPARSE ENCODER?

Using a model trained with Deformable-DETR, we have analyzed the number of encoder output tokens referenced by the decoder's object query. Unlike using the bilinear interpolation described in the appendix A.2 to generate DAMs for training with pseudo-labels, in this analysis, we do not use bilinear interpolation to calculate how many encoder tokens are directly referenced by the decoder object query. To analyze the non-zero values of DAM, we use a Deforamble DETR model trained with Top-$k$ sampling strategy (Yao et al., 2021) and bounding box refinement (Zhu et al., 2021) using ResNet-50 backbone. Fig. 10 illustrates the distribution of the ratio of non-zero values of DAM on COCO `val2017` dataset. As shown in Fig. 10, on average, only 45% of encoder tokens were referenced by object queries.

This observation naturally raises a question: Can we preserve the detection performance even if we focus, in the first place, only on the encoder tokens that the decoder might have preferred? As a preliminary experiment to answer this question, we trained the detector restricting token updates to the subset to which the decoder could have referred if there had been no such restriction. To this end, we performed the two-stage learning as follows: (i) We first obtained the DAMs of the entire

training data by feeding them to a fully-trained Deformable-DETR model. (ii) Then, we retrained another model from the scratch by updating only a subset of tokens determined by the binarized DAM preserving top-$\rho\%$ of the elements(refer to Section3.3 for more details). Table 7 shows the performance on COCO detection for different keeping ratio $\rho$. We found that the two-stage model almost catches up with the baseline($\rho = 100\%$) as the keeping ratio is raised close to 45%, namely the percentage of non-zero values in DAM computed on the validation dataset earlier.

These observations have strongly motivated us to develop the encoder token sparsification method presented in the main text. Note that our main algorithm differs from this preliminary experiment in some aspects: (a) A DAM is obtained from the jointly learning decoder, not from the separately trained decoder, and (b) a binarized DAM is utilized as a prediction target of the scoring network rather than used directly as a sparsification mask.

### A.10 VISUALIZATIONS OF SELECTED ENCODER TOKENS

We visualize selected encoder tokens and top-$k$ decoder queries for each criterion, OS, and DAM. In the first row, selected encoder tokens from the backbone feature map are visualized as yellow regions, whereas unselected tokens are visualized as purple regions. In the second row, selected top-$k$ decoder queries from encoder output are visualized in the same manner. In the final row, DAM values and $Corr$ metrics are visualized. $Corr$ is measured as in Section 4.2.

Interestingly, the DAM-based selection seems to better capture the objects than OS-based selection. The OS-based selection also captures objects well, but it typically focuses on the high-frequency edges that are not only in the foreground but also in the background. On the other hand, the DAM-based selection captures the boundary of the objects and also their inner areas and is less distracted from the background edges. We analyze that DAM focuses on the boundary of objects to lower the regression loss, and attends to the inside of objects to lower the classification loss. Finally, the scoring network predicts such a DAM well, and refining the encoder tokens according to it finally helps achieve better detection performance.

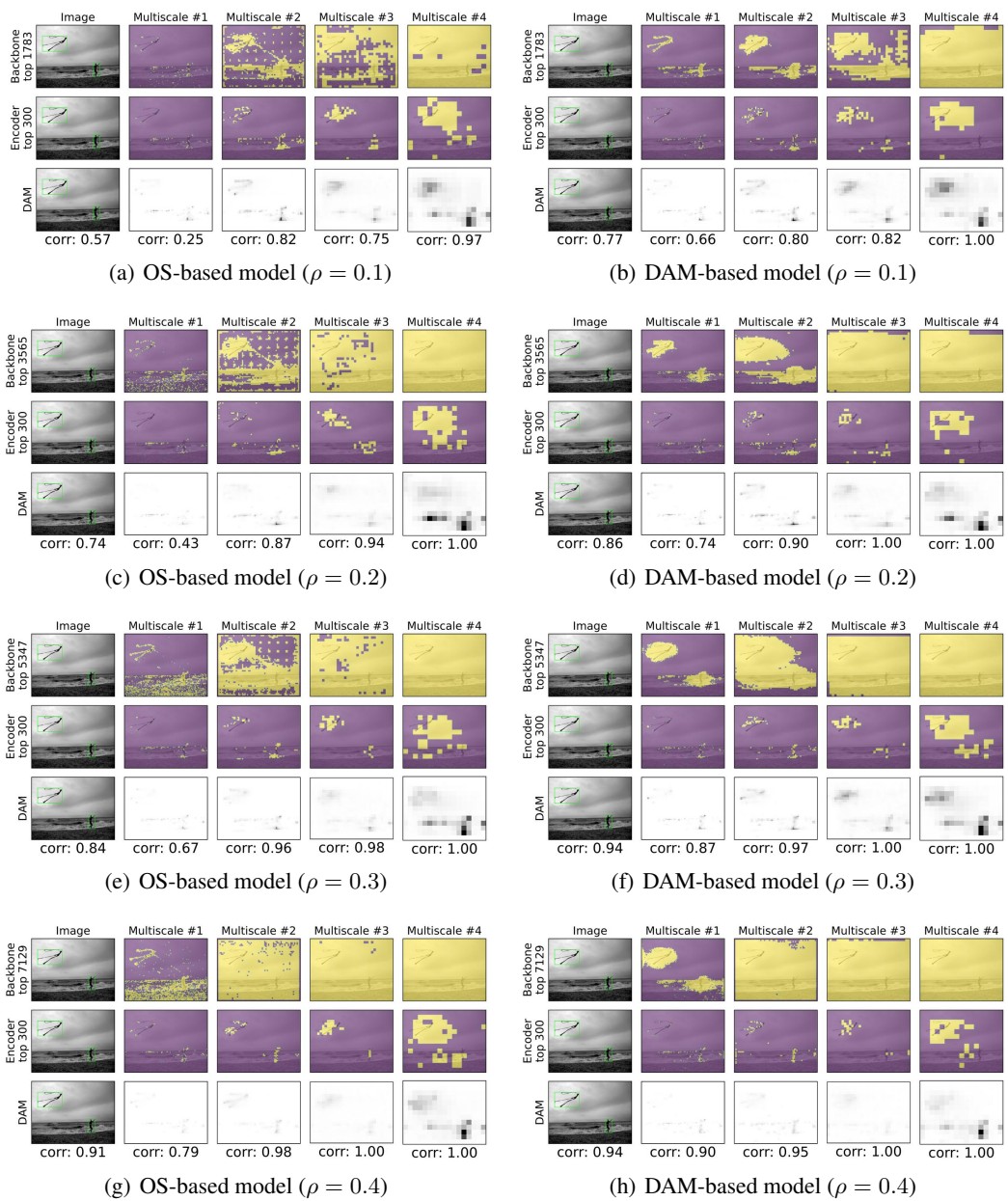

Figure 11: Visualization of selected tokens and DAM for COCO validation image #289960

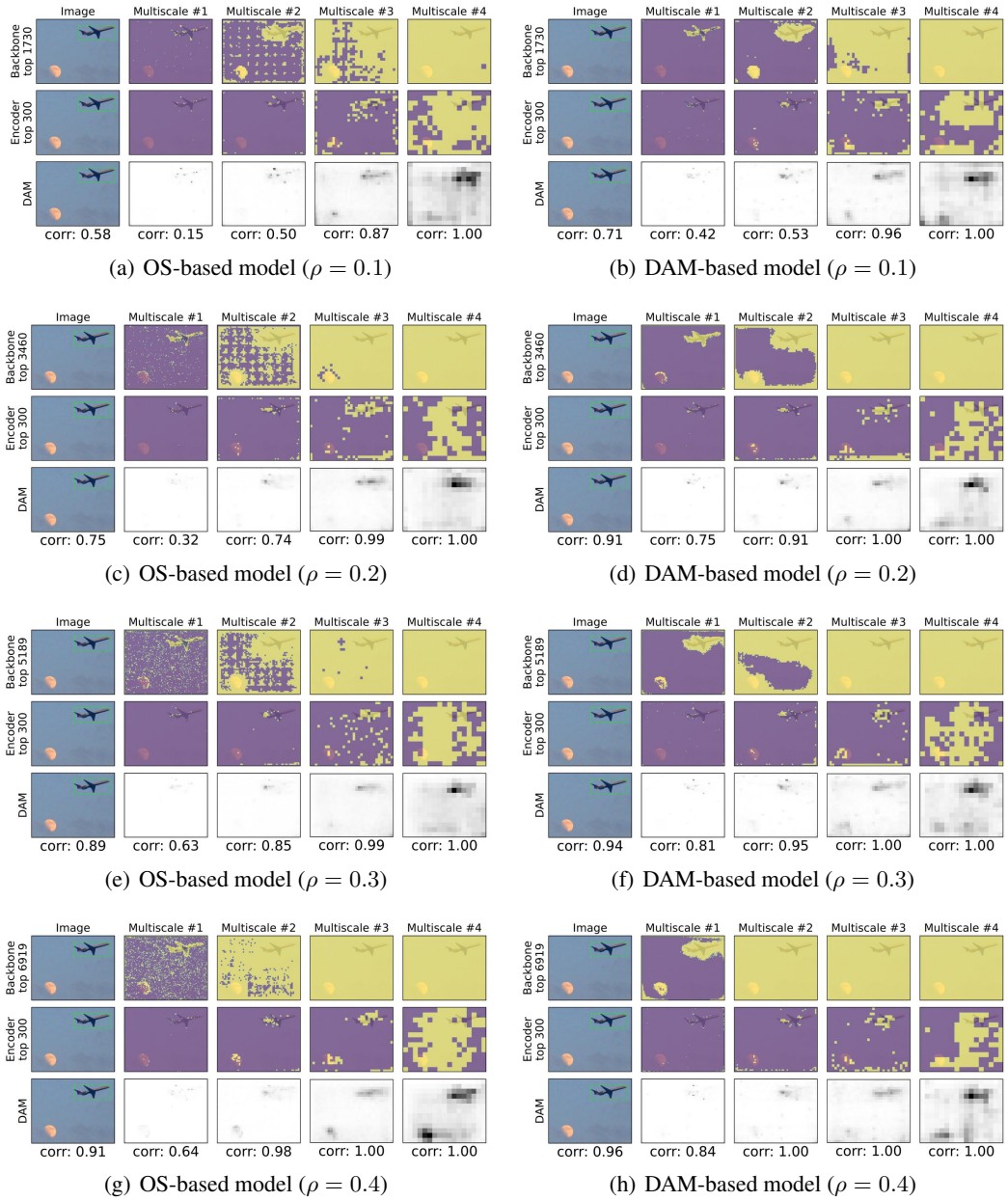

Figure 12: Visualization of selected tokens and DAM for COCO validation image #22396

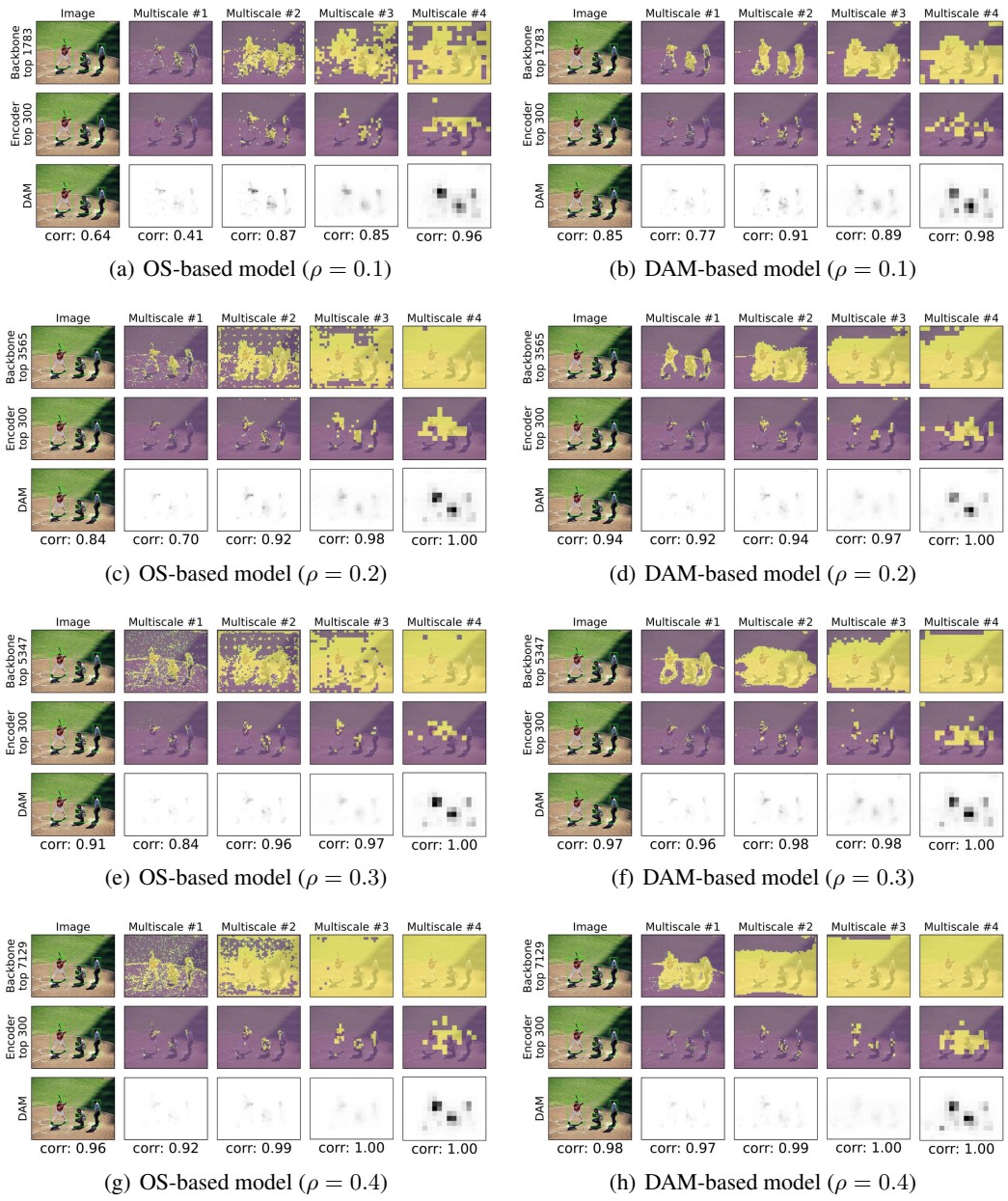

Figure 13: Visualization of selected tokens and DAM for COCO validation image #46252

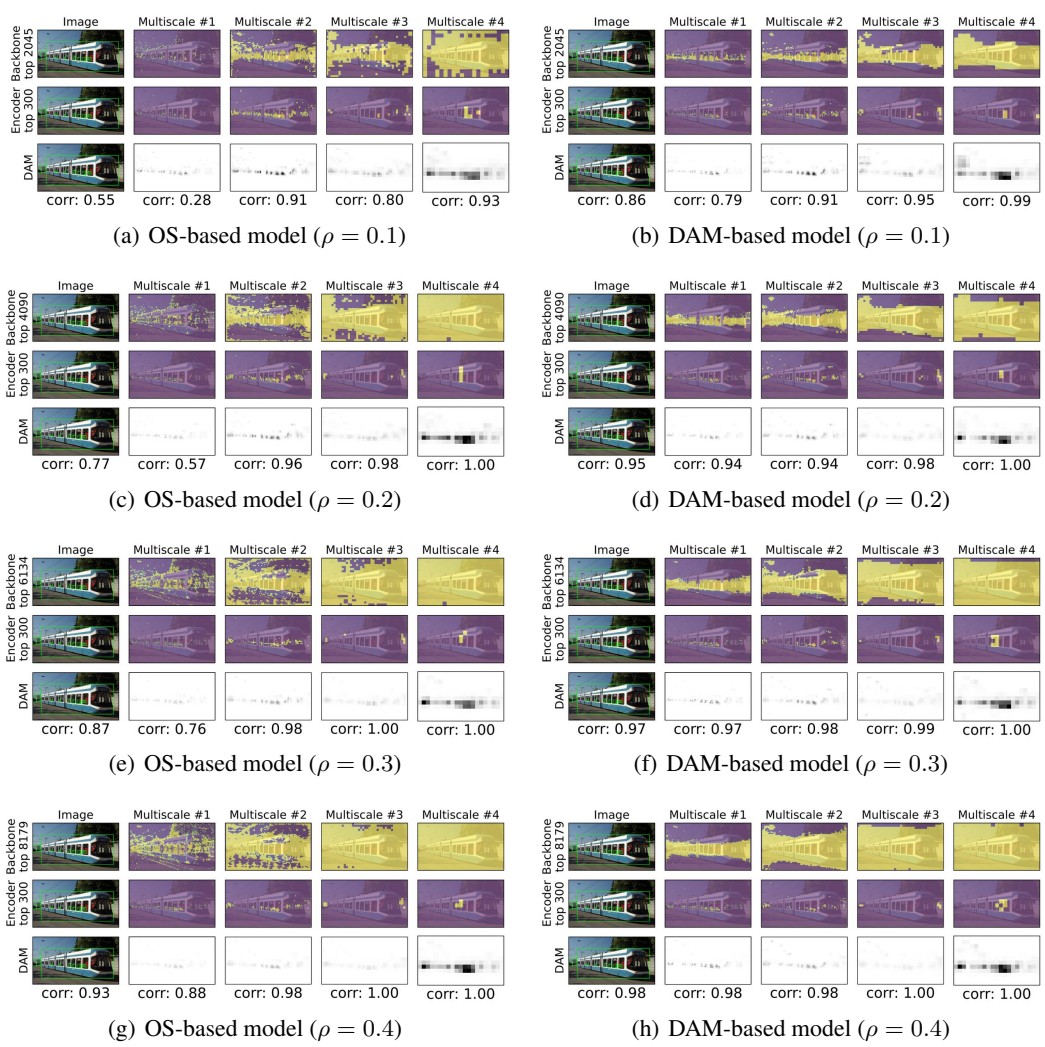

Figure 14: Visualization of selected tokens and DAM for COCO validation image #6040

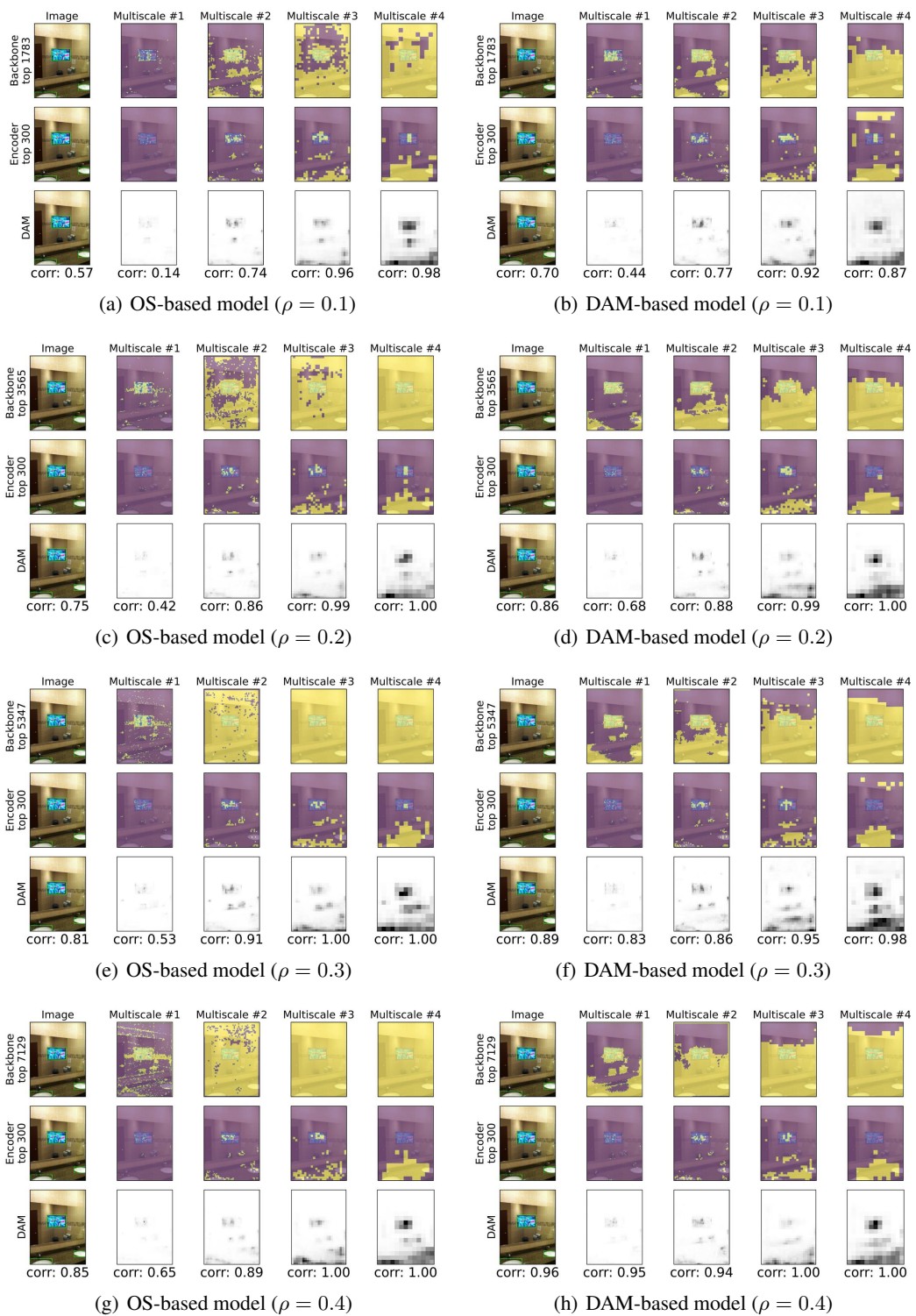

Figure 15: Visualization of selected tokens and DAM for COCO validation image #17379

