# OpenReview forum: "Sparse DETR: Efficient End-to-End Object Detection with Learnable Sparsity"
_ICLR.cc/2022/Conference — ICLR 2022 Poster_

### Official Review · Reviewer_jT45 · 2021-11-01

**Correctness:** 3
**Technical Novelty And Significance:** 3
**Empirical Novelty And Significance:** 3
**Recommendation:** 6
**Confidence:** 5

**Main Review:**

Strengths:
The method is novel and technically sound. As we all know that the queries or tokens are somewhat redundant for DETR-based object detection, this work brings us some new insight of how to efficiently condense queries without performance dropping. Compared to PnP and other dynamic ViTs, this work is more effective and efficient. The writing of this work is also clear and easy to follow.

Weakness:
In table 1, the 50% ratio Sparse DETR is higher (46.3 vs 46.0) than Deformabel DETR under the same training setting. In my understanding, the full queries version of Deformabel DETR is equal to a 100% ratio version of sparse DETR. If so, why the performance is even higher when we only keep 50% of queries. I think this result is deserved for more analysis. Or are there any additional components that can improve the total performance of the full version of the detector?


**Summary Of The Paper:**

This paper proposes an efficient end-to-end object detection architecture based on Deformable DETR, called Sparse DETR. Its main contribution is solving the computation budget by selecting only a few tokens as queries in attention computing. Sparse DETR can use only 10%-50% of the original encoder query while achieving comparable results. This is done by several components. Firstly, the authors add a subnet named Scoring Net before encoder to predict saliency scores for input. And only the tokens which have top-k scores will be chosen as queries in the encoder attention computing. Then, to train the Scoring Network, this paper use binarized attention weight from cross attention, named DAM, as pseudo labels. It also adds a detection head to predict the class scores for outputs of the encoder. The scores are used to select the appropriate token as the decoder`s query. The extensive experiments show the effectiveness of the proposed method.

**Summary Of The Review:**

Overall, this paper provides valuable insight into reducing queries for DETR-based detectors and I prefer to accept it.

---

> ### Author Response · Authors · 2021-11-19
> **Response to Reviewer jT45**
>
>
>
> We really appreciate your constructive comments. We respond to each comment as follows.
>
> > **Comment:** In table 1, the 50% ratio Sparse DETR is higher (46.3 vs 46.0) than Deformable DETR under the same training setting.
>
> **Response**
>
> Deformable DETR in **Table 1** is slightly different with a 100% ratio version of Sparse DETR, in that Sparse DETR uses additional components (*the proposed encoder auxiliary loss*. please refer to **Section 3.4**).
>
> Comparison with the full query is described in **Figure 4**. Sparse DETR of keeping ratio 50% performs slightly worse than 100% version. Also, **Figure 6** describes the effectiveness of the proposed encoder auxiliary loss: (1) it enhances detection performance and (2) stabilizes the learning process when with more encoder layers (3) extra computation is only marginal, thanks to encoder token sparsification.
>
> For clarity, we have added the explanation to the revised paper.

---

### Official Review · Reviewer_YsVT · 2021-11-02

**Correctness:** 3
**Technical Novelty And Significance:** 4
**Empirical Novelty And Significance:** 4
**Recommendation:** 8
**Confidence:** 5

**Main Review:**

Overall the method seems fairly simple and should be straightforward to implement. The sparisification contributions are well supported experimentally, with ablations and insights.

Contribution #3, the encoder auxiliary loss, is not well supported. The original DETR paper shows that the encoder can be scaled to 12 layers and show improved performance over 6 layers (+1AP), table 2. An additional investigation on what could be the reason behind convergence issues in deep Deformable DETR is required. Training deep Transformer encoder models is extensively studied in NLP, with well known practices like BERT init and pre-norm block structure. The contribution needs to be backed up by convincing evidence on why it would be preferable over the common practices.

It would be beneficial to show results with instance or panoptic segmentation, eq with SOLQ which was build on top of Deformable DETR as well. Would in this case more topk features be required to match performance, would be proposed method still be a useful method to improve computational efficiency? Would Objectness Score be a better sparsification target than DAM in case of masks?

Suggestions to improve writing:
- In table 1, results for R50 with SCRL init could be moved to appendix as they are not directly support the claims.
- The first sentence of the abstract where the authors say "DETR demonstrates competitive performance but low computational efficiency" is confusing, since in table 1 DETR actually has the lowest FLOP count/fastest runtime. I guess what the authors meant was that "DETR demonstrates competitive performance but low computational efficiency on high resolution feature maps". Overall the abstract needs to be significantly improved, it has undefined terms and is very difficult to read. For example, in the sentence "Using the multiscale feature to ameliorate performance.." neither multiscale nor feature is mentioned before.
- Figures 4-7 lack captions explaining what the reader is supposed to understand from them.
- Figures 2-3 would benefit from highlighting inputs and outputs
- The authors use word "polish" when describing Transformer layers applied to feature maps, which is slightly confusing. Perhaps there is a better word for it.
- notation $x_\text{bb}$ is confusing, since $\text{bb}$ typically means "bounding box", and not "backbone".
- Section 3.1: "decoder takes both the refined queries ... " Transformer encoder output is typically called "memory", it would disambiguate encoder output from object queries.



**Summary Of The Paper:**

A modification to the recently proposed Transformer-based object detector, DETR, is proposed. The authors are motivated by low performance of DETR on small objects, and high computational cost of self-attention on large feature maps. To improve both, they propose to sparsify input feature maps by learning a classifier whether to include an input feature or not. Supervision for this classifier comes from Transformer decoder attention weights. The authors apply their approach to Deformable DETR and evaluate on the COCO bounding box detection benchmark, and show that it allows to trade detection performance for computational efficiency by tuning the classifier threshold.

**Summary Of The Review:**

The paper suggests a novel approach for improving computational efficiency of Transformer-based object detectors. Most contributions are supported by experiments, ablations and insights. The contribution with auxiliary losses in encoder requires more investigation. Overall I recommend accept.

---

> ### Author Response · Authors · 2021-11-19
> **Response to Reviewer YsVT**
>
>
> We really appreciate your constructive comments. We respond to each comment as follows.
>
> ---
>
> > **Comment:** Contribution #3, the encoder auxiliary loss, is not well supported.
>
> **Response**
>
> As you pointed out, the original DETR reported that the encoder can be scaled to 12 layers. However, we failed to reproduce this result using the official code with only the number of encoder layers varied. Interestingly, we also observed that the model is trained successfully with the ‘*pre-norm*’ option. On the other hand, Deformable DETR fails to converge even with the ‘*pre-norm*’ technique, while the encoder auxiliary loss certainly facilitates stable convergence.
>
> We assumed that the vanishing gradient problem underlies this observation and the proposed encoder auxiliary loss may remedy this issue. To verify this, we visualized the gradient norm of each layer in encoders of different depths. We found that the vanishing gradient effect actually occurs for any size of encoder but more heavily as it is deeper. Besides, we observed that the ‘*pre-norm*’ for DETR and the ‘*encoder auxiliary loss*’ for Deformable DETR actually keeps the gradient in the early layer significantly large, which allowed the stable training even when the number of encoder layers increased.
>
> We have added the new section in the revised manuscript, **Appendix A.5**, where you can find details on these experiments.
>
> ---
>
> > **Comment:** It would be beneficial to show results with instance or panoptic segmentation.
>
> **Response**
>
> Unfortunately, we could not conduct a segmentation experiment due to the time constraint. However, we believe that ours will also perform well on segmentation task, since 1) as shown in **Figure 11~14**, visualization of selected encoder tokens raises expectations for segmentation tasks, in the sense that the model mainly selects the interior of the object despite training using bounding box annotations only, and 2) PnP DETR, similar encoder token sparsification approach, also performs well in the segmentation task.
>
> ---
>
> > **Comment:** Would in this case more topk features be required to match performance? Would the proposed method still be a useful method to improve computational efficiency?
>
> **Response**
>
> We believe that a similar number of top-$k$ features would be required, thus it will help to improve efficiency. Visualization of selected encoder tokens (in **Appendix A.10**) shows that the model of keeping ratio 50% captures the boundary of objects and some of backgrounds, which might mean that about 50% of the tokens can demarcate the boundary of objects.
>
> ---
>
> > **Comment:** Would Objectness Score be a better sparsification target than DAM in case of masks?
>
> **Response**
>
> Even in segmentation tasks, we believe that DAM would be a better target. In the visualization of selected encoder tokens (in **Appendix A.10**), the model using DAM better captures the boundary of objects than the model using OS. It even captures the boundaries of objects that are not included in the detection classes.
>
> ---
>
> > **Comment:** Suggestions to improve writing
>
> **Response**
>
> Thanks for the constructive suggestions. We have reflected the comments in the revised manuscript.

---

### Official Review · Reviewer_sTnV · 2021-11-03

**Correctness:** 3
**Technical Novelty And Significance:** 2
**Empirical Novelty And Significance:** 2
**Recommendation:** 5
**Confidence:** 5

**Main Review:**

-The motivation is interesting somehow. The computational and memory complexity of Deformable DETR are indeed some problems. It would be interesting to reduce its complexity but remain accuracy.

-The techniques to achieve the goal, e.g. adding a saliency head or using the decoder saliency map, are kind of empirical and less interesting.

-I am quite confused about the two additional components in Sec. 3.4. I don’t get their motivations and they seem to be irrelevant to the main motivation of this paper. What are their effects? Don’t see detailed ablations on them in the experiments.

-The improvements on both accuracy and speeds are reasonably good, but not very impressive.

-The writing/presentation needs to be improved. There are quite a few confusing/unclear points in the paper, which are difficult to understand.


Some detailed comments:

-in the abstract, it says only 45% queries are referred and not updating them don’t drop the accuracy. How is this implemented? A very naïve baseline? Don’t see it in the experiments.

-The auxiliary loss was first used in [A], instead of GoogleNet.

-For the query selection, once the queries are selected, they are active for all encoders or each encoder has its own selection?

-For the inactive queries, are they inactive for all encoders or they can be inactive for 1st encoder but active for the 2nd?

-For the inactive queries, aren't they not involved in the computation graph at all? how could they be referenced when updating the selected queries?

-Fig. 2 is quite difficult to read.

-For DAM, if you already use the decoder for saliency, how can you not compute the encoder? Aren’t the encoder’s outputs the decoder’s inputs?

-Why need another 4-layer scoring network? Don’t you already use decoder to compute the saliency?

-How to add auxiliary loss to encoder? There are no object queries as input to the encoder, so no detections for encoder.

-In table 1, what’s the deformable detr with a check mark? And what’s Deformable DETR+?


[A] Deeply-Supervised Nets, AISTATS 2015

**Summary Of The Paper:**

This paper tries to solve the problem of expensive computation on the encoder of Deformable DETR. The hypothesis is the encoder inputs a large number of image feature queries, but only a small number of them are actually referred by the decoder. The paper has shown that with a help with some selection mechanism, the encoder and only inputs partial of the feature queries, such that the computation can be reduced. With some other components, e.g. “top-k decoder queries” and “encoder auxiliary loss”, the Sparse DETR can achieve slightly better performances at fewer computations.

**Summary Of The Review:**

The motivation is somehow interesting; the techniques are kind of empirical and less interesting; the improvements over Deformable DETR are not very surprising on both accuracy and speeds.

---

> ### Author Response · Authors · 2021-11-19
> **Response to Reviewer sTnV (2/2)**
>
> > **Comment:** For DAM, if you already use the decoder for saliency, how can you not compute the encoder? Aren’t the encoder’s outputs the decoder’s inputs? Why need another 4-layer scoring network? Don’t you already use decoder to compute the saliency?
>
> **Response**
>
> The role of the scoring network is to predict saliency of tokens from the feature map, since the decoder attention map is not accessible right after the backbone network. For clarity, we briefly summarize the whole process in *forward* and *backward* passes:
> - *In the forward pass*, three steps are sequentially executed; 1) the scoring network predicts which tokens are important, 2) the top-$\rho$ % tokens are selected based on this importance, and 3) these tokens are selectively updated, and passed to the decoder. For more information, please see **Section 3.2**.
> - *In the backward pass*, we compute the binarized DAM and train the scoring network to predict DAM. For more details, please refer to **Section 3.3**.
>
> This whole process is illustrated in **Figure 2**.
>
> ---
>
> > **Comment:**  How to add auxiliary loss to encoder? There are no object queries as input to the encoder, so no detections for encoder.
>
> **Response**
>
> To implement the encoder auxiliary losses, we treat encoder tokens as object queries and attach the detection head along with Hungarian matching loss on the top of them. Please refer to **Section 3.4**, ‘*Encoder Auxiliary Loss*’ paragraph.
>
> Classifying and regressing all pre-defined anchors corresponding to feature maps in RPN of Faster RCNN or SSD [1] is a well-known method in object detection frameworks. Based on these well-known methods in object detection, we treat all encoder tokens as object queries and apply Hungarian loss to each token. Note that, in Sparse DETR, only part of encoder tokens are refined by the encoder, and adding auxiliary heads only for sparsified encoder tokens is not a big burden compared to dense one.
>
> ---
>
> > **Comment:**  In table 1, what’s the deformable detr with a check mark? And what’s Deformable DETR+?
>
> **Response**
>
> This check mark means that the model is trained with top-$k$ strategy and bounding box refinement. We denote Deformable DETR with the top-$k$ object query selection and bounding box refinement, as Deformable DETR+. Please refer to **Section 4.1**.
>
>
> ---
>
> **Reference**
>
> [1] W. Liu et al. SSD: Single Shot MultiBox Detector. ECCV 2016

---

> ### Author Response · Authors · 2021-11-19
> **Response to Reviewer sTnV (1/2)**
>
> We really appreciate your constructive comments. We respond to each comment as follows.
>
> ---
>
> > **Comment:** The techniques to achieve the goal, e.g. adding a saliency head or using the decoder saliency map, *are kind of empirical and less interesting.*
>
> **Response**
>
> We still believe that using decoder attention maps to train the scoring network is a reasonable (and interesting) solution to sparsify the encoder tokens, since it is explicitly designed to select encoder tokens that are very relevant to the decoder. In addition, we want to emphasize that our approach has been developed on intriguing observations:
>
> 1. We observe that only 45% of encoder tokens were referenced by decoder object queries. For the details on how to obtain this statistic, please refer to the response to Reviewer u6Zb or **Appendix A.9**, in the revised manuscript.
>
> 2. We see that only updating the selected encoder tokens do not degrade the performance too much, since gradients from the decoder are propagated to the encoder only through the referenced encoder tokens.
>
> ---
>
> > **Comment:**  I am quite confused about the two additional components in Sec. 3.4. *I don’t get their motivations and they seem to be irrelevant to the main motivation* of this paper. What are their effects? Don’t see detailed ablations on them in the experiments.
>
> **Response**
>
> These additional components (*top-$k$ decoder queries* and *encoder auxiliary loss*) have been introduced to improve the final performance and stabilize the optimization.
>
> The manuscript contains an ablation study on the *encoder auxiliary loss* in **Section 4.3** and **Figure 6**, showing the improvement of detection performance and stabilizing the learning process, without sacrificing too much computational cost. Note that, adding the encoder auxiliary loss to the conventional DETR and Deformable DETR will be a computational burden, due to the many encoder tokens and therefore *encoder auxiliary loss* is also relevant to encoder query sparsification.
>
> Also, we have added **Appendix A.5** that describes additional experiments of the proposed *encoder auxiliary loss*: (1) the existence of vanishing gradient in the deep encoders (2) and alleviation of vanishing gradient with the proposed *encoder auxiliary loss*.
>
> On the other hand, we omitted an ablation study on the *top-$k$ decoder query selection*, since this has been validated in Efficient DETR.
>
> ---
>
> > **Comment:**  In the abstract, it says only 45% queries are referred and not updating them don’t drop the accuracy. *How is this implemented?* A very naïve baseline? Don’t see it in the experiments.
>
> **Response**
>
> To be precise, there have been two different preliminary experiments and the statement you have mentioned is relevant to both of them. We can summarize them as follows:
> - The one is to survey how many encoder tokens are actually referred by the decoder. This experiment was performed with the fully-trained Deformable-DETR model on COCO validation set. From this experiment, we found that the fully-trained detector, in nature, focuses on a portion(45%) of the encoder tokens.
> - The other one is to see whether the detector preserves its performance if we have focused only on the preferable encoder tokens from the beginning of the training. In this experiment, we trained the detector from scratch on the COCO training set, while narrowing the token update in the encoder to the precomputed DAM from another fully-trained detector. From this experiment, we found that developing an encoder sparsification method is a feasible direction.
>
> We have revised the manuscript to avoid confusion between the above two tasks and devoted another whole section, **Appendix A.9**, for much more details.
>
> ---
>
> > **Comment:**  The auxiliary loss was first used in *Deeply-Supervised Nets*, instead of *GoogleNet*
>
> **Response**
>
> We added the paper to related works. Thanks for letting us know.
>
> ---
>
> > **Comment:**   For the query selection, once the queries are selected, *they are active for all encoders or each encoder has its own selection?* For the inactive queries, are they inactive for all encoders or they can be inactive for 1st encoder but active for the 2nd?
>
> **Response**
>
> Once the queries (i.e. the encoder tokens) are active/inactive, these are active/inactive for all encoders. We use the term "encoder" to refer to the entire encoder layers.
>
> ---
>
> > **Comment:**   For the inactive queries, aren't they not involved in the computation graph at all? how could they be referenced when updating the selected queries?
>
> **Response**
>
> We remark the inactive queries can be still referenced by a deformable attention module in each encoder layer, thereby participating in the computation graph. In Eq. (1), $\text{DefAttn}$ both takes $x_{i-1}^{j}$ (selected tokens) and $x_{i-1}$ (all tokens), meaning that inactive queries are also involved in the graph.
>
> ---
>
> (more on below)

---

### Official Review · Reviewer_u6Zb · 2021-11-03

**Correctness:** 4
**Technical Novelty And Significance:** 3
**Empirical Novelty And Significance:** 3
**Recommendation:** 8
**Confidence:** 5

**Main Review:**

Overall, I think this is a good paper that has

a. reasonable motivations and backgrounds

b. reasonable and effective solutions

c. supportive benchmark results and sufficient ablation studies

d. good written and easy to follow.

It focuses on a specific problem in current DETR based methods and proposes a reasonable solution/direction to further improve the efficiency of the model.

There are some suggestions to improve the literature:

a. There is an argument that “the encoder queries referenced by the decoder account for only 45% of the total”. However, I do not observe any statistical analysis and support in the main paper. A more detailed analysis is recommended to add to better support the paper.

b. The paper uses the word “encoder queries”. It doesn’t have problems. However, the encoder queries (i.e. backbone features) are key/values in the decoder, not the queries. In DETR and Deformable DETR, they all use the word “feature maps”, because from the perspective of the entire Transformer, the query is the learnable object embeddings. I recommend the authors revise this part to make the presentation more clear.


**Summary Of The Paper:**

This paper proposes methods to further improve the efficiency of Deformable DETR. It observes that the encoder queries referenced by the decoder account for only 45% of the total, and the detection accuracy does not deteriorate significantly even if only the referenced queries are polished in the encoder block. To this end, this paper proposes Sparse DETR that selectively updates only the queries expected to be referenced by the decoder. An auxiliary detection loss in the encoder also improves the performance. Experiments prove the effectiveness of the method.

**Summary Of The Review:**

Overall, this is a good paper that has good motivations and reasonable solutions. The experiments are thorough. I recommend to accept the paper and hope the authors can revise the paper according to my suggestions.

---

> ### Author Response · Authors · 2021-11-19
> **Response to Reviewer u6Zb**
>
>
> We really appreciate your constructive comments. We respond to each comment as follows.
>
> ---
>
> > **Comment:** Statistical analysis of argument “*the encoder queries referenced by the decoder account for only 45% of the total*”
>
> **Response**
>
> To analyze the number of encoder tokens referenced by decoder’s object query, we first trained Deformable DETR on ResNet-50 backbone with top-$k$ sampling strategy and bounding box refinement. Using the trained Deformable DETR model, we have observed that only 45% of encoder tokens were referenced by decoder object queries on average over COCO validation samples. And, this is a good reference to the reason why sparsifying encoder tokens does not hurt the performance. More details on this observation and statistical analysis have been newly added in **Appendix A.9** with a **Figure 10**.
>
>
>
> ---
>
> > **Comment:** Term “*encoder queries*”
>
> **Response**
>
> Thanks for the comment for clarity. We’ve changed the term “*encoder queries*” into “*encoder tokens*” to prevent confusion. In relation to this, all sentences in the manuscript that can be interpreted vaguely have been changed.

---

### Author Response · Authors · 2021-11-19
**A Summary of Paper Updates**

Thanks to all reviewers for constructive suggestions that help make this work more complete.
Following their suggestions, we have made the following major updates to the manuscript, including **+3 pages** in Appendix, to provide more justifications for the paper:

- *Sentences that may have ambiguous meanings have been corrected throughout the paper.*
- **Section 3**: Figure 2 and Figure 3 have been newly revised.
- **Section 4**: The results for ResNet-50 with SCRL initialization in Table 1 have been moved to Appendix A.7.
- **_(New!)_ Appendix A.5**: We have analyzed that the vanishing gradient problem occurs when the number of encoder layers increases, and that the proposed encoder auxiliary loss helps alleviate this problem.
- **_(New!)_ Appendix A.9**: We have visualized the distribution of the ratio of non-zero values of DAM, explained the preliminary experiments that became the motivation for our method, and reported the results.

---

Please refer to the paper for further details.

---

### Decision · Program_Chairs · 2022-01-20

**Decision:**

Accept (Poster)

**Comment:**

This paper proposes to modify DETR, a recent Transformer-based architecture for object detection.
More precisely,  they propose to sparsify input feature maps by learning an extra classifier to select which input features (few of them) will be used in the attention module. The supervision of this classifier is guided by second extra module  coming from the Transformer decoder attention weights.
The resulting framework, called Sparse DETR, is an efficient end-to-end object detection architecture that allows to overcome the main computational bottleneck of DETR. Sparse DETR can use  only 10%-50% of the original encoder query while achieving DETR comparable results.

Authors tried to answer to all the questions raised during the rebuttal.
Even if the final scores are still contrasted, most of reviewers are very positive.
Overall, this paper provides valuable insight into reducing computational complexity (by reducing the number of queries) for DETR-like detectors. The proposed method is novel and technically sound. Even if there are some tricks to make the whole working well, this work is more effective and efficient than previous propositions to handle this complexity problem,  bringing us some new insight of how to sparsify queries without performance dropping.
All these elements lead me to propose this paper for publication at ICLR.